# Sensing Utilities of Cesium Lead Halide Perovskites and Composites: A Comprehensive Review

**DOI:** 10.3390/s24082504

**Published:** 2024-04-13

**Authors:** Muthaiah Shellaiah, Kien Wen Sun, Natesan Thirumalaivasan, Mayank Bhushan, Arumugam Murugan

**Affiliations:** 1Department of Research and Analytics, Saveetha Dental College and Hospitals, Saveetha Institute of Medical and Technical Sciences (SIMATS), Saveetha University, Chennai 600077, India; muthaiahs.sdc@saveetha.com (M.S.); mayankbhushan.sdc@saveetha.com (M.B.); 2Department of Applied Chemistry, National Yang-Ming Chiao Tung University, Hsinchu 300, Taiwan; 3Department of Periodontics, Saveetha Dental College and Hospitals, Saveetha Institute of Medical and Technical Sciences (SIMATS), Saveetha University, Chennai 600077, Tamil Nadu, India; natesant.sdc@saveetha.com; 4Department of Chemistry, North Eastern Regional Institute of Science & Technology, Nirjuli, Itanagar 791109, India; amu@nerist.ac.in

**Keywords:** cesium lead halides, analyte detection, environmental monitoring, cellular imaging, fluorescent quantification, chemical sensors, real analysis, pesticides, herbicides, photodetection

## Abstract

Recently, the utilization of metal halide perovskites in sensing and their application in environmental studies have reached a new height. Among the different metal halide perovskites, cesium lead halide perovskites (CsPbX_3_; X = Cl, Br, and I) and composites have attracted great interest in sensing applications owing to their exceptional optoelectronic properties. Most CsPbX_3_ nanostructures and composites possess great structural stability, luminescence, and electrical properties for developing distinct optical and photonic devices. When exposed to light, heat, and water, CsPbX_3_ and composites can display stable sensing utilities. Many CsPbX_3_ and composites have been reported as probes in the detection of diverse analytes, such as metal ions, anions, important chemical species, humidity, temperature, radiation photodetection, and so forth. So far, the sensing studies of metal halide perovskites covering all metallic and organic–inorganic perovskites have already been reviewed in many studies. Nevertheless, a detailed review of the sensing utilities of CsPbX_3_ and composites could be helpful for researchers who are looking for innovative designs using these nanomaterials. Herein, we deliver a thorough review of the sensing utilities of CsPbX_3_ and composites, in the quantitation of metal ions, anions, chemicals, explosives, bioanalytes, pesticides, fungicides, cellular imaging, volatile organic compounds (VOCs), toxic gases, humidity, temperature, radiation, and photodetection. Furthermore, this review also covers the synthetic pathways, design requirements, advantages, limitations, and future directions for this material.

## 1. Introduction

To protect the ecosystem, the detection, quantification, and removal of environmental contaminants play a vital role [1,2,3]. Thus, the synthesis and fabrication of novel nanomaterials and nanocomposites are important in developing various analytical methods [4,5,6,7,8]. Metal halide perovskites (including all inorganic and organic–inorganic perovskites) and their composites, in particular, have been demonstrated as unique probes in diverse analyte quantitation [9,10]. However, most halide perovskites suffer from chemical/structural instability caused by exposure to moisture, oxygen, and high temperature [9,10,11]. To resolve the instability problem, the development of all inorganic perovskites was proposed to make them sustainable under harsh conditions [12]. Furthermore, the use of all inorganic halide perovskites (such as inorganic metal oxides, lead-free metal halides, and cesium lead halides) in analyte detection displayed exceptional performance in real-time applications [13]. Among all inorganic perovskites, cesium lead halides (CsPbX_3_; X = Cl, Br and I) and composites have been widely demonstrated in photovoltaic applications and in the detection/quantification of metal ions, anions, chemicals, explosives, bioanalytes, pesticides, fungicides, cellular imaging, volatile organic compounds (VOCs), toxic gases, humidity, temperature, and radiation [14,15,16,17,18,19].

The distinct sensor responses of CsPbX_3_ can be attributed to its excellent electro-optical properties, which benefit the development of numerous semiconducting and sensing utilities [18,19]. For instance, CsPbX_3_ displays high chemical stability at higher temperatures (at >350 °C) and exhibits bright emission with high PLQY reaching >90% (PLQY= photoluminescence quantum yield) [20]. The bandgap of CsPbX_3_ (lies between 1.7 and 3 eV) can be adjusted across the visible spectrum by tuning the X-site ion and composition ratios to attain red-to-blue emission [21,22,23]. Depending on the temperature and the size of halide (X) ions, CsPbX_3_ could possess different crystal phases, such as cubic (Pm-3m, α), tetragonal (P4/mbm, β), and orthorhombic (Pnam γ and Pnma δ (noted as non-perovskite) phases [24,25]. Likewise, the surface morphology and potentials may vary when absorbing diverse analytes (moisture, gaseous species, environmental contaminants, bioanalytes, etc.) [26,27,28]. Variations in photoluminescence/absorption, phase transformation, and changes in the surface morphology and charge potentials of CsPbX_3_ can be considered as sensor responses [29,30,31,32,33] when detecting analytes. These sensors can be further enhanced by combining perovskites with suitable/proper nanomaterials [34,35,36,37,38,39,40,41,42,43].

The electro-optical properties and sensor responses of CsPbX_3_ and composites may vary depending on their distinct nanostructures. For example, CsPbX_3_-based sensor probes/composites in QD structures with various sizes may possess diverse bandgaps and display red-to-blue wide optical properties [34], which facilitates the design of dual-mode sensors. Subsequently, CsPbX_3_ nanocrystals (NCs) also display unique magnetic and optoelectronic properties. The facile synthesis of NCs allows them to be adopted in distinct applications, such as solar cells and in vitro/in vivo applications [35]. Due to their structural features, such as hardness, diffusivity, density, enhanced ductility/toughness, elasticity, and conductivity/thermal properties, CsPbX_3_ NCs can be effectively applied in energy-related studies. For example, Hu et al. defined the use of CsPbBr_3_ NCs as single-photon emitters [36]. Metal nanoclusters (MNCs) showed exceptional physicochemical properties, such as surface modifiability, surface-to-volume ratio, number of atoms, biocompatibility, photothermal stability, etc. [37]. Therefore, conjugating with CsPbX_3_ may enhance the performance of the target-specific sensors. Because the reduced dimensionality of nanowires (NWs) can significantly improve electric/heat transport compared with bulk wires, they have great potential as temperature and chemoresistive sensors [38,39]. For instance, Zhai and co-workers reported the solvothermal synthesis of CsPbX_3_ (X = Cl, Br) NWs and demonstrated them in photodetector applications [40]. Regarding two-dimensional materials, nanosheets (NSs) have been demonstrated as effective sensors due to their exceptional physical, chemical, optical, mechanical, electronic, and magnetic properties [41]. Lv et al. demonstrated the generalized colloidal synthesis of two-dimensional cesium lead halide perovskite nanosheets in photodetector applications [42]. Furthermore, nanoparticles with high surface-to-volume ratios were employed in multiple sensors, which can be operated at distinct solvent environments and elevated temperatures [43]. Based on the above reasons, CsPbX_3_ probes/composites derived from QDs, NCs, MNCs, NWs, NSs, and NPs require detailed review.

The exceptional optical properties, unique structural/crystalline features, and electronic structures of CsPbX_3_ (X = Cl, Br, and I) are considered important material properties for electrochemical, thermal, and chemoresistive sensing studies. To date, numerous optical sensors made of CsPbX_3_ (X = Cl, Br, and I) and composites have been thoroughly investigated with exceptional applicability [9,10,11,12,13,14,15]. This can be attributed to their distinct and high PLQY in red-to-blue luminescence. However, there have been reports on the electrochemical, thermal, and chemoresistive sensing performance of CsPbX_3_ (X = Cl, Br, and I) and composites [9,10,11,12,13,14,15] that require further clarification for future research. Heavy metal ions and anions are well-known environmental contaminants that are involved in cellular processes, and, at high concentrations, they may become harmful to living beings as well [44,45,46]. Chemicals and explosives also contaminate the environment; thus, their detection methods are available in many reports [47,48,49]. Toxic gases and VOCs are noted as vital industrial contaminants; thus, their quantitation has been explored by numerous researchers [50,51]. Exposure to radiation, temperature, and high humidity may harm living tissues and beings, and therefore researchers have developed sensors for photo, radiation, and photodetection [52,53,54]. Bioanalytes, drugs, fungicides, and pesticides play crucial roles in food cycles and sustain the living environment; thus, numerous reports are available for their identification [55,56,57,58]. Based on the aforementioned important issues, many researchers have adopted CsPbX_3_ (X = Cl, Br, and I) and composites for the optical, electrochemical, chemoresistive, and thermal detection of analytes. The progress and challenges in developing these sensors are reviewed in this article.

Numerous sensors are reported that involve the use of CsPbX_3_ and composites toward the detection of metal ions, anions, chemicals, explosives, bioanalytes, pesticides, fungicides, cellular imaging, VOCs, toxic gases, humidity, temperature, X-rays, and photons (light). Many reviews covering halide perovskite-based sensors, including hybrid halide perovskites and all inorganic halide perovskites, are available [9,10,59,60,61,62,63,64]. However, most reviews do not provide much detail in sensor studies of CsPbX_3_-based composites and the underlying sensor mechanisms. Therefore, the focus of this article is to review the sensing utilities of CsPbX_3_ and composites toward diverse analytes (see Figure 1) and provide valuable information on the synthetic pathways, design requirements, advantages, limitations, and future directions.

## 2. Role of Structural Stability and Optoelectronic Properties in Sensors

CsPbX_3_ crystals following the stoichiometry ABX_3_ have an undistorted cubic structure composed of Pb^2+^ surrounded octahedrally by ‘X’ anions (X = Cl, Br, and I) and a larger Cs^+^ cation with a 12-fold cuboctahedral coordination [65]. CsPbX_3_ in a cubic crystal structure can be stable only if the values of the Goldschmidt tolerance factor are between 0.9 and 1 [66]. The Goldschmidt tolerance factor of CsPbX_3_ is determined from t = (R_A_ + R_X_)/√2 (R_B_ + R_X_), where R_A_, R_B_, and R_X_ are the ionic radii of Cs^+^, Pb^2+^, and halide (X) ions, respectively. When the values of the tolerance factor lie between 0.7 and 0.9, the CsPbX_3_ crystal may exist in distorted cubic structures with variations in symmetries/phases [65,66,67]. The structural distortion and instability of CsPbX_3_ may arise from external factors, such as temperature, quantum size moisture, etc. [68,69]. For example, phase transitions among cubic, monoclinic, tetragonal, or orthorhombic phases occur in the CsPbX_3_ crystal at a temperature range between 300 and 600 K [70]. Similar to the variations in particle sizes and compositions, the phase transformation in the CsPbX_3_ may also result in broad emission covering the entire visible range (from blue to red), which can be utilized to develop analyte sensors. For example, Protesescu et al. [71] demonstrated CsPbX_3_ NCs with tunable bandgap energies by controlling the quantum size and colloidal compositions of NCs, as shown in Figure 2. When exposed to moisture, humidity, gaseous environment, and the doping of foreign materials, structural distortions via surface-mediated absorption, oxidation, reduction, etc., could occur in CsPbX_3_, which can be regarded as sensor responses. The stability of hybrid halide perovskites also follows a similar trend as CsPbX_3_-based materials [9]. However, metal oxide perovskites show slightly better stability than that of CsPbX_3_ and hybrid halide perovskites [9,10,11,12].

Owing to its exceptional optoelectronic properties, CsPbX_3_ can be applied in analyte quantification by monitoring its responses in photoluminescence, absorbance, conductivity, temperature, etc. Moreover, CsPbX_3_ possesses bright blue-to-red emission, depending on the halide (X) concentration, and displays high PLQY (can reach over 90%). Therefore, CsPbX_3_ and composites can also be utilized in the quantification of halides [72]. CsPbX_3_ and composites can display unique absorbance and colorimetric responses in the presence/absence of specific analytes [73]. The density functional theory calculations conducted by Y. Kang and co-workers [74] showed that different halide (X) ions in CsPbX_3_ can lead to changes in intrinsic carrier mobility by a factor of 3 to 5, depending on the carrier concentration, which is between 10^15^ and 10^18^ cm^−3^. Their work also concluded that, in terms of carrier mobility, the preferred carrier type (electron or hole) also depends on halide (X). Kawano et al. investigated the halogen ion dependence on the low thermal conductivity of cesium halide perovskites using first-principle phonon calculations [75]. E. G. Ripka and co-workers reported variations in surface-ligand binding potential due to the halide ion exchange to afford diverse emissions and PLQY [76]. The changes in intrinsic carrier mobility, thermal conductivity, and surface-ligand binding potential can be adopted in designing various electrochemical, temperature, and colloidal sensors for the detection of specific analytes. However, some of the reported sensors were attributed to a combination of carrier mobility and surface-ligand binding. For instance, exposing CsPbX_3_ to gaseous or VOC analytes resulted in variations in conductivity, but the underlying mechanism was attributed to the efficacy of surface-ligand binding in the modification of oxidation or reduction [77,78,79].

## 3. CsPbX_3_ (X = Cl, Br, and I) and Composites toward Metal Ion Detection

Due to the importance of heavy metal ion detection in environmental protection, the use of CsPbX_3_ (X = Cl, Br, and I) and composites in metal ion sensors have been widely reported by many researchers, which are described in this section. Wu and co-workers demonstrated the PL-based detection of Cu^2+^ in the presence of ytterbium acetate (Yb(OAc)_3_) by engaging one-dimensional (1D)-CsPbCl_3_ NCs in the device [80]. The CsPbCl_3_ NCs were synthesized using the hot-injection method and characterized by transmission electron microscopy (TEM), X-ray diffraction (XRD), X-ray photoelectron spectroscopy (XPS), and Fourier transform infrared spectroscopy (FTIR). Yb(OAc)_3_ induces morphology changes from weakly emissive 1D CsPbCl_3_ NCs (PLQY = 2.1%) to highly luminescent 1D CsPbCl_3_ NWs (PLQY = 17.3%), which are low in defect density and high in conductivity. The PLQY depends on the size of nanostructures (due to bandgap variations), which can be adjusted with metal doping. The doping of Yb^3+^ in 1D-CsPbCl_3_ NWs resulted in a higher aspect ratio, uniformity, and lower number of defects compared with undoped ones; thus, Yb-doped 1D-CsPbCl_3_ NWs showed high PLQYs. The enhanced defective and rough surface morphologies of 1D-CsPbCl_3_ NCs effectively hindered light absorption and electron/charge transport, thereby lowering the PLQY. In the presence of acetate ion (AcO^−^; present in Yb(OAc)_3_), the steric hindrance was reduced by the copper-based counter-ion pair, which enhanced the adsorption of Cu^2+^ on the surface of the CsPbCl_3_ NCs and resulted in PL quenching. The linear regression of Cu^2+^ detection was observed between 0 and 1 µM (µM = micromole (10^−6^ M)) with an LOD of 0.06 nM (nM = nanomole (10^−9^ M)). This work is informative and reveals much detail on the role of Yb(OAc)_3_, underlying the dynamic quenching mechanism, surface-mediated analyte interaction, and feasible information of electron transfer/charge trap between CsPbCl_3_ and Cu^2+^. However, it lacks real-time applications, which should be demonstrated before commercialization.

Sheng et al. reported the use of CsPbBr_3_ QDs in the quantification of Yb^3+^ and Cu^2+^ [81]. CsPbBr_3_ QDs were synthesized using the hot-injection method, with a PLQY of 63%, and showed enhancement in PL with Yb^3+^ and PL quenching with Cu^2+^, as seen in Figure 3. The linear PL quenching range of Cu^2+^ is between 2 nM and 2 µM, with an estimated LOD of 2 nM. This work is impressive in reporting Cu^2+^ detection in edible oils but lacks detailed investigations on the underlying mechanisms. Liu et al. also described the use of CsPbBr_3_ QDs (PLQY = 90%) toward the luminescent detection of Cu^2+^ [82]. CsPbBr_3_ QDs were synthesized via the hot-injection method and displayed PL quenching between 0 and 100 nM, with an LOD of 0.1 nM. The detection of Cu^2+^ was carried out in organic media hexane following a dynamic quenching mechanism. This is a follow-up work of previous reports with additional validation on selectivity and time-resolved studies.

To avoid using organic media in sensor investigations, Kar and co-workers reported that the use of poly(vinyl pyrrolidone), n-isopropylacrylamide-coated CsPbBr_3_ NCs (PVP-NIPAM-CsPbBr_3_ NCs) for aqueous media facilitated the discrimination of Cu^2+^ [83]. Firstly, SiO_2_-coated CsPbBr_3_ NCs and PVP-NIPAM-CsPbBr_3_ NCs were synthesized via ligand–ligand-assisted reprecipitation (LARP) method to achieve a maximum PLQY of 93%. The PVP-NIPAM-CsPbBr_3_ NCs showed improved stability and dispersity in water compared to silica-coated NCs, with redshifted emission peaks around 513–515 nm. The NCs displayed linear PL quenching between 0 and 412 µM, with a calculated LOD of 18.6 µM, as seen in Figure 4. This is an inspiring work that can be readily extended to biological studies. Li et al. reported the use of green-fluorescent CsPbBr_3_ (CPB) QDs for Cu^2+^ quantification using phase transfer [84]. A strong organic ligand (oleylamine, OAm) was added to selectively transfer Cu^2+^ from water to cyclohexane, which led to fluorescent quenching. The PL emission was quenched linearly between 1 µM and 10 mM. This is an innovative method that enables Cu^2+^ detection via phase transfer with a short response time (1 min). However, no real applications were reported in this work. Moreover, Song and co-workers developed long-wavelength-pass filters consisting of CsPbBr_3_ and CsPb(Cl/Br)_3_ QDs using a Cu^2+^-quenching strategy [85]. However, no clear real-time applications were demonstrated in this work.

Li et al. proposed the utilization of macroporous CsPbBr_3_-(SH)polyHIPE NCs for the ultrasensitive detection of Cu^2+^ via PL quenching [86]. In this work, CsPbBr_3_ NCs were synthesized via the hot-injection method, with a PLQY of ~98%. The as-synthesized NCs were then composited with (SH)polyHIPE (generated from the monomers trimethylolpropane triacrylate (TMPTA) and trimethylolpropane tris(3-mercaptopropionate) (TMPTMP)). The linear regression of PL quenching was recorded between 10 fM and 10 mM, with an LOD of 10 fM. Although this work is supported by density functional theory (DFT) calculations, it lacks real-time applications. Gao and co-workers developed a CsPbBr_3_ QD-based fluorescence-enhanced microfluidic sensor for the in situ detection of Cu^2+^ in lubricating oil [87]. As displayed in Figure 5a, the polymethyl methacrylate opal photonic crystal (PMMA OPC) film is a microfluidic sensor substrate, which displays high sensitivity and low LOD with Cu^2+^. When coupling with OPCs, CsPbBr_3_ QDs synthesized through the hot-injection method showed a 26-fold enhancement in PL intensity (at 496–526 nm under 450 nm excitation). When adding Cu^2+^ in lubricating oil, the PL intensity of PMMA OPC/CsPbBr_3_ QD composites was quenched with Cu^2+^ concentrations of 1 nM–10 mM and an LOD of 0.4 nM, as shown in Figure 5b. The CsPbI_3_ QD/SiO_2_ IOPC (inverse opal photonic crystal) composite was further explored for the on-site rapid detection of Cu^2+^ [88]. CsPbI_3_ QDs synthesized with a hot-injection method displayed PL emission at 693 nm under 405 nm excitation. SiO_2_ IOPCs were introduced into the chip wells to couple with the CsPbI_3_ QDs, which further enhanced the PL emission by 22 folds. After the addition of Cu^2+^ in lubricating oil to the above composites, a linear PL quenching was observed in the ranges of 0–20 nM and 20–50 nM, with an LOD of 0.34 nM. Both reports [87,88] show distinct performance in terms of mechanical durability, operating temperatures (5 °C to 50 °C), reusability of chips, etc. Advancing in this research direction will require further demonstrations of practical applications.

Zhang and co-workers proposed the utilization of organic cross-linker hexamethylene diisocyanate (HDI)-reinforced small-sized CsPbBr_3_@SiO_2_-E NPs (size ≤ 50 nm) toward the fluorescent sequential detection of Cu^2+^ and S^2−^ [89]. CsPbBr_3_ NCs (size = 6.8 nm; emission at 515 nm) were first synthesized using a hot-injection technique and then conjugated via a three-step synthetic path to afford CsPbBr_3_@SiO_2_-E NPs (emission peak at 508 nm; PLQY = 90%), where ‘E’ stands for enhanced performance. The linear PL quenching/recovery of CsPbBr_3_@SiO_2_-E NPs with Cu^2+^ and S^2−^ were observed in ranges of 0–5 µM and 5–10 µM (for Cu^2+^) and 0–120 µM (for S^2−^), with corresponding LODs of 0.16 µM (for Cu^2+^) and 8.8 µM (for S^2−^). This work requires further support with real-time investigations. The CsPbBr_3_ QDs (PLQY = 88%) were encapsulated in polymethyl methacrylate (PMMA) fiber membrane (d ≈ 400 nm) to afford CPBQD/PMMA FM for detecting trypsin, Cu^2+^, and pH [90]. The CPBQD/PMMA FM and cyclam interacted effectively to capture Cu^2+^, with a linear range of 1 fM–1 M (fM = femtomole (10^−15^ M)) and an LOD of 1 fM. Moreover, fluorescence resonance energy transfer (FRET) between CPBQD and Cu^2+^ plays a vital role, resulting in PL quenching. Trypsin detection by CPBQD/PMMA FM via PL quenching was attained in the presence of peptide CF6 (Cys–Pro–Arg–Gly–R6G). Similarly, CPBQD/PMMA FM and R6G were combined to display pH-mediated fluorescent quenching. This is a unique work with exceptional sensor investigations. However, it can be improved further by supporting additional real-time applications. Ahmed et al. proposed a two-step surfactant-free procedure for producing a CsPbBr_3_ QD-embedded zinc(II) imidazole-4,5-dicarboxylate metal–organic framework (MOF) for the luminescent detection of Cu^2+^ [91]. The PL emission of CsPbBr_3_@MOF composites at 519 nm (under 360 nm excitation; PLQY = 39.2%) was quenched linearly between 100 and 600 nM, with an estimated LOD of 63 nM. PL quenching occurred through dynamic quenching and electron transfer with a Stern–Volmer quenching constant (K_SV_) of 1.55 × 10^5^ M^−1^. Although this report is an impressive work, it lacks real-sample investigation.

Wang et al. reported the use of a liquid–liquid extraction technique for the visual detection of Hg^2+^ in aqueous media [92]. In their study, the luminescent CsPbBr_3_ NCs (PL emission at 520 nm under 380 nm excitation) were synthesized using the hot-injection method to engage in the liquid–liquid-extraction-based visual detection of Hg^2+^. When adding Hg^2+^ dissolved in water into CsPbBr_3_ NCs in carbon tetrachloride (CCl_4_), the colorimetric PL emission quenching at 520 nm via liquid–liquid extraction was observed, as displayed in Figure 6. The linear regression of Hg^2+^ was recorded between 50 nM and 10 µM, with an estimated LOD of 35.65 nM. This work requires further investigations into the underlying mechanism and real-time applications.

Jiang et al. proposed a two-step precipitation method to synthesize emissive CsPbBr_3_ crystals (PL emission at 525 nm under 395 nm excitation) toward Hg^2+^ detection [93]. When detecting Hg^2+^, both CsPbBr_3_ NCs and Hg^2+^ were co-precipitated in aqueous solution. The CsPbBr_3_ precursor was firstly dissolved in an aqueous solution containing Hg^2+^ (0–1000 nM) and then dropped onto a hydrophilic polydimethylsiloxane (PDMS) substrate with a microwell array. When the substrate was heated at 25 °C for 3 min, co-precipitation occurred, which resulted in PL quenching via Hg^2+^ doping into the CsPbBr_3_ lattice. The linear regression of Hg^2+^ was observed between 5 and 100 nM, with an estimated LOD of 0.1 nM. This work demonstrated an innovative technique and provided detailed studies on interference, pH effect, and underlying mechanisms toward Hg^2+^ detection. However, the cost-effectiveness and real-time applications of this method require more work. Through ligand engineering and silica encapsulation, a stable fluorescent CsPbBr_3_-mPEG@SiO_2_ composite (PL emission at 520 nm under 330 nm excitation; PLQY = 67.5%) was synthesized and adopted in the sequential detection of Hg^2+^ and glutathione (GSH) in aqueous solution via PL quenching and recovery, respectively [94]. Shu and co-workers demonstrated that the existence of 73% of the PL emission of NCs could last over 30 days in aqueous media. The PL quenching and recovery responses were attributed to the electron transfer process between NCs and Hg^2+^ and the effective interaction between Hg^2+^ and GSH. The linear PL responses of Hg^2+^ and GSH were observed in the ranges of 1–50 nM and 1–10 µM, with LODs of 0.08 nM and 0.19 µM, respectively. This work was successfully applied in tab and serum sample analysis; therefore, it can be regarded as a remarkable work in Hg^2+^ and GSH detection.

Guo et al. developed a nucleation growth method for producing CsPbBr_3_ NCs (PL emission at 518 nm under >360 nm excitation; PLQY > 89%) at a large scale and adopted as-synthesized NCs for detecting Zn^2+^ [95]. The PL emission of CsPbBr_3_ NCs was quenched linearly between 0 and 40 µM in the presence of Zn^2+^. The PL quenching was not due to the replacement of Pb^2+^ in the CsPbBr_3_ matrix but was caused by the Zn–oleic complex formation. The surface defects created led to the self-assembly of CsPbBr_3_ nanocubes into nanorods, thereby resulting in PL quenching. George and co-workers reported the use of alpha-amino butyric acid (A-ABA)-capped CsPbBr_3_ QDs (M PQDs) for developing Co^2+^ sensors [96]. The M PQDs were synthesized using the hot-injection method, which displayed PL emission at 489 nm. The PL emission of the M PQDs was quenched in Co^2+^ concentrations of 0–100 nM, with an LOD of 0.8 µM. This report uncovered that PL quenching was due to FRET-facilitated dynamic quenching and the inner filter effect (IFE). This is the only report on IFE-based PQD sensors using metal ions, but it lacks information in real-time applications. Halali et al. reported uranyl (UO_2_^2+^) ion detection using green emissive CsPbBr_3_ PQDs (synthesized using the hot-injection method) [97]. When adding UO_2_^2+^, the PL emission at 518 nm was quenched linearly in UO_2_^2+^ concentrations of 0–3.3 µM, with a calculated LOD of 83.33 nM. Extensive mechanistic studies revealed that the PL quenching was due to the electrostatic interaction and adsorption of UO_2_^2+^ over the surface of QDs. To support this work, further research on the interference and application studies is necessary.

Polyvinylpyrrolidone (PVP) polymer shell-grown silica-coated Zn-doped CsPbBr_3_ NCs (polymer-coated Zn-doped CsPbBr_3_/SiO_2_ core/shell NCs (PVP-0 NCs, PVP-2.5 NCs, PVP-5 NCs, PVP-7.5 NCs, and PVP-10 NCs)) were synthesized with the hot-injection method for detecting In^3+^ in water [98]. The double-coating method enhanced the water stability, dispersibility, and emission properties of the NCs. Among the various PVP shell-grown silica-coated Zn-doped CsPbBr_3_ NCs, PVP-5 NCs (PLQY = 88%) were more stable at higher temperatures and showed stronger luminescence and greater selectivity to In^3+^. When adding In^3+^, the PL emission of the PVP-5 NCs at 511 nm was quenched in In^3+^ concentrations of 0–104 µM, with an estimated LOD of 11 µM. The PL quenching was associated with the replacement of Pb^2+^ by In^3+^. This report is noteworthy but lacks evidence for the proposed mechanism and real-time applications. Pandey et al. employed the CsPbBr_3_−Ti_3_C_2_T_x_ MXene QD/QD heterojunction for the PL-based detection of Cd^2+^ [99]. CsPbBr_3_ QDs were synthesized using the hot-injection method and then were composited with Ti_3_C_2_T_x_ MXene in toluene. The PL emission of CsPbBr_3_−Ti_3_C_2_T_x_ MXene QDs at 505 nm (under 410 nm excitation) was quenched via charge transfer when adding Cd^2+^. The linear PL quenching of the QD composite was observed in Cd^2+^ concentrations of 99–590 µM with no information on the value of LOD. This work also demonstrated an on−off−on PL probe for cadmium ion detection, but more investigations are necessary to justify the underlying quenching (static/dynamic) mechanisms. Hsieh and co-workers proposed the use of (3-aminopropyl) triethoxysilane (APTES)0coated CsPbBr_3_–CsPb_2_Br_5_ QDs toward the PL-based detection of Fe^3+^, as illustrated in Figure 7 [100]. Through the ligand-assisted reprecipitation method, APTES-coated CsPbBr_3_–CsPb_2_Br_5_ QDs were synthesized. The PL emission of the QDs at 520 nm was quenched rapidly (response time = 8 s at 40 °C) in the presence of Fe^3+^. The linear PL responses of QDs to Fe^3+^ were observed in Fe^3+^ concentrations of 10 µM–10 mM with an LOD of 10 µM. This is a well-organized work with excellent results in response time and temperature, but it lacks supportive data on mechanistic investigations. Table 1 summarizes the synthetic route, PLQY, linear range, detection limit, and application of CsPbX_3_ (X = Cl, Br, and I) and composites for metal ion detection.

### Critical Comments on CsPbX_3_ (X = Cl, Br, and I)-Based Metal Ion Detection

Based on the existing results, it is noted that as-synthesized CsPbX_3_ QDs, NCs, and NWs display high specific selectivity to Cu^2+^ through feasible energy transfer between the probes and Cu^2+^ [80,81,82]. Furthermore, it was clarified that Yb^3+^ doping enhanced the selectivity by reducing the surface defect [80,81], thereby suggesting the effectiveness of surface forces in sensors. Another critical issue in the use of CsPbX_3_ (X = Cl, Br, and I) probes for metal ion detection, which requires more attention, is their stability in aquatic environments. To solve the stability issues, using polymer and ligand capping/coating on CsPbX_3_ has been proposed [83,84,86,89,90], which may enhance the PLQY by avoiding surface exposure to environmental forces existing in water and air. However, whether this approach can be effective in exposure to Cu^2+^ in an aquatic environment remains an open question. The development of a pass filter consisting of CsPbBr_3_ and CsPb(Cl/Br)_3_ QDs was demonstrated for Cu^2+^ detection via PL quenching responses [85]. However, the development of such pass filters has not met commercial standards. It is a premature proposal and requires additional work. CsPbBr_3_ QD/CsPbI_3_ QDs were explored using the microfluidic technique, which facilitated the detection of Cu^2+^ [87,88]. However, the fabrication processes of such devices are rather complicated and require a well-equipped clean room environment, thereby restricting their advancement in most developing countries. Also, it is essential to determine whether this microfluidic method is effective in all environmental samples. The use of CsPbBr_3_ crystals, NCs, and QDs was also reported in the PL “turn-off” detection of Hg^2+^ and UO_2_^2+^ [92,93,94,97]. Many of the available reports on CsPbX_3_ (X = Cl, Br, and I)-based metal ion sensors confirmed their selectivity to Cu^2+^; however, the underlying mechanisms of detecting Hg^2+^ and UO_2_^2+^ by CsPbBr_3_ crystals, NCs, and QDs are still unclear. Likewise, the composites of CsPbX_3_ (X = Cl, Br, and I) with other emerging nanomaterials, such as MOFs, Mxene, APTES, etc., have been proved to be effective in discriminating diverse heavy metal ions [91,94,95,96,98,99,100]. However, most of those reports did not address the feasible surface-facilitated detection mechanisms, which restricted the development of analytical devices. These results also raise the question of the reliability of CsPbX_3_ (X = Cl, Br, and I)-based Cu^2+^ sensors. The reason behind the selective sensing of Cu^2+^ must be clarified by investigating the Pb^2+^ replacement mechanism, as well as the magnetic property (ferro-/ferri-electronic) changes. The crystalline and lattice features of the probes/compositions in the presence/absence of analytes are not considered from mechanistic aspects, which should be taken into account in future sensor designs. If the crystalline/lattice parameters of CsPbX_3_ (X = Cl, Br, and I)-based probes are taken into account, it is highly feasible to design chemoresistive and electrochemical sensors for heavy metal quantification in real samples.

## 4. Anion Detection by CsPbX_3_ (X = Cl, Br, and I) and Composites

Similar to metal ion quantification, the discrimination of anions was also demonstrated by perovskite nanomaterials, as described in this section. Jan et al. reported the synthesis of the CsPbBr_3_ nanoplatelets (PLQY = 83.7%) via the hot-injection method, which displayed PL emission at above 475 nm under 350 nm excitation [101]. The PL peak was blueshifted when the CsPbBr_3_ nanoplatelets were exposed to Cl^−^ (from a HCl source). The sensor response showed a linear range from 0.2 to 0.4 nM, with an LOD of 28 pM. The observed response was attributed to the anion exchange mechanism. Moreover, CsPbBr_3_ nanoplatelets are also able to effectively detect the arsenate in the presence of hypochlorous acid (HOCl). The following reaction process (1) shows that As^3+^ is oxidized to produce Cl^−^:AsO_3_^3−^ + OCl^−^ → AsO_4_^3−^ + Cl^−^(1)

A blueshifting phenomenon in PL emission from the reaction induced by Cl^−^ species occurred via anion exchange, which showed a linear regression of 6.4–58 nM, with an LOD of 1 nM. This is an innovative work with dual-species recognition; however, it does not provide enough real-time applications. Thereafter, Huang et al. proposed the use of CsPbBr_3_ QDs for Cl^−^ detection in water [102]. The CsPbBr_3_ QDs, with PL emission at 513 nm and a calculated PLQY value of 87%, were synthesized using the hot-injection method. When exposed to Cl^−^, the PL emission peak at 513 nm was blueshifted to 483 nm due to the anion exchange reaction with Br^−^. The shifting of the PL peak occurred in Cl^−^ concentrations of 10–200 μM, with an estimated LOD of 4 µM. This report was well supported by real-time water analysis; therefore, it can be regarded as a unique work. Shu and co-workers developed highly stable CsPbBr_3_ NCs via amphiphilic polymer ligand-assisted synthesis [103]. Amphiphilic polymer octylamine-modified polyacrylic acid (OPA) was used as the capping agent to produce stable NCs, with PL emission at 520 nm and >40% PLQY. As displayed in Figure 8, the PL peak at 520 nm is blueshifted to 441 nm due to the anion exchange reaction in Cl^−^ concentrations of 1–80 mM, with an LOD of 0.34 mM. This report demonstrated Cl^−^ detection in sweat samples; thus, it is an innovative work. However, further research work is required for commercialization.

Shortly after, Li et al. adopted CsPbBr_3_ NCs for the luminescent colorimetric sensing of Cl^−^ in n-hexane via a halide exchange reaction [104]. The CsPbBr_3_ NCs were synthesized using the hot-injection method, in which a rapid halide exchange reaction occurred at pH = 1. The green emissive peak of the CsPbBr_3_ NCs at 514 nm was blueshifted to a 452 nm peak (blue emission) in Cl^−^ concentrations of 10–130 mM, with an estimated LOD of 3 mM. This work was applied in Cl^−^ detection in sweat samples; hence, it is quite innovative. However, the values of the LOD must be further improved by combining other techniques. By taking advantage of the anion exchangeability of CsPbBr_3_ NCs, Dutt and co-workers proposed the construction of a glass plate-/paper-strip-based test kit for discriminating Cl^−^ [105]. The PL emission peak of the kit at 509 nm was slowly blueshifted to 478 nm because of the anion exchange reaction. The linear regression of Cl^−^ was observed between 100 µM and 10 mM with an LOD of 100 µM. This work requires more supportive evidence for possible commercialization.

Recently, Zhang and co-workers developed β-cyclodextrin (β-CD)-stabilized, arginine (Arg)-added CsPbBr_3_ NCs (ACD-PNCs; PLQY = 82%) via ligand-assisted synthesis and utilized them in discriminating Cl^−^ and I^−^ through a ligand exchange mechanism [106]. β-CD capping, together with the addition of Arg, helped to stabilize the PNCs. The green emission of ACD-PNCs was blue/redshifted from 508 nm to 424 nm (blue emission) and 511 nm to 637 nm (red emission), respectively, in the presence of Cl^−^ and I^−^. The linear regression of Cl^−^ and I^−^ detection was observed in the ranges of 0.04–0.8 mM and 0.04–1.16 mM, with calculated LODs of 3.2 µM and 9 µM, respectively. This is a unique work that provides a comparative study with earlier reports. However, perspectives on further work are not mentioned. As discussed in many studies related to anion exchange reactions, CsPBBr_3_ can act as exceptional probes toward the quantification of Cl^−^/HCl, I^−^, and F^−^ [107,108]. An alcohol-dispersed CsPbBr_3_@SiO_2_ PNCC nanocomposite was proposed for discriminating Cl^−^ in an aqueous phase [109]. The green emissive peak at 506 nm was blueshifted to 447 nm through the homogeneous halide exchange between CsPbBr_3_@SiO_2_ PNCCs and Cl^−^. The recovery studies of Cl^−^ in sea sand samples (with a linear range of 0–3%) attested to the reliability of this work, with an LOD of 0.05 mg/g. Moreover, the anion exchange between CsPbBr_3_@SiO_2_ PNCCs and Cl^−^ occurred in the absence of magnetic stirring or pH regulation, which was a novel observation.

Fu et al. proposed the use of the NH_2_-functionalized CsPbBr_3_ NCs for detecting I^−^ [110]. These NH_2_-functionalized NCs were synthesized in ethanol by using 3-aminopropyltriethoxysilane (APTES) as ligands. In contrast to the traditional halide exchange-based I^−^ sensors of CsPbBr_3_, the luminescence of the NH_2_-PNCs in ethanol/water at 510 nm was quenched, as shown in Figure 9a. Linear regression is observed in I^−^ concentrations of 4–28 µM, with an LOD of 1 µM, as seen in Figure 9b. I^−^ showed higher selectivity among all other interferences, but this report lacks supporting evidence on the PL quenching response of the unshifted peak.

Park and co-workers fabricated a CsPbBr_3_ QD/cellulose composite as an early diagnosis sensor for Cl^−^ and I^−^ [111]. The CsPbBr_3_ QD/cellulose composite was synthesized via a hot-injection method to form monodispersed CsPbBr_3_ QDs with high selectivity to Cl^−^ and I^−^. The detection of Cl^−^ and I^−^ in aqueous media was confirmed by observing a color change from green to blue and from green to red, respectively. The color change occurred within 5 s because of the halide exchange reaction. The linear responses of the CsPbBr_3_ QD/cellulose composite to Cl^−^ and I^−^ were recorded at 0.1 mM–1 M, with calculated LODs of 2.56 mM (for Cl^−^) and 4.11 mM (for I^−^), respectively. In terms of real-time applications, this work can be regarded as a unique report on medical device fabrication. The halogen ion exchange reaction of CsPbBr_3_ and composites brings an additional advantage of the effective discrimination of edible oils. Zhang et al. demonstrated the discrimination of edible oils by using octadecylammonium iodide (ODAI) and ZnI_2_ as anion exchangers [112]. This colorimetric sensing strategy can be applied in detecting edible oil mixtures with 100% accuracy, but further research is required for commercialization. Wang et al. described the employment of tetraphenylporphyrin (TPPS)-modified CsPbBr_3_ NCs (CsPbBr_3_/TPPS nanocomposite) for the quantification of sulfide (S^2−^) [113]. The CsPbBr_3_/TPPS nanocomposite possessed good water stability and dual-emission properties. As shown in Figure 10, the CsPbBr_3_/TPPS nanocomposite displays strong green emission at 520 nm and moderate red emission from the TPPS at 650 nm. When adding S^2−^, the PL emission at 520 nm was quenched linearly in S^2−^ concentrations of 0.2–15 nM, with an LOD of 0.05 nM. This was attributed to the destruction of CsPbBr_3_ NCs via the formation of more stable PbS. This work also reported the real-time water recovery study (>95%), which showed a relative standard deviation of <3%, thereby opening a new direction for future research. Table 2 summarizes the synthetic route, PLQY, linear range, LOD, and application of CsPbX_3_ (X = Cl, Br, and I) and composites toward the detection of anionic species.

### Critical View on CsPbX_3_ (X = Cl, Br, and I)-Based Anion Sensors

CsPbX_3_ (X = Cl, Br, and I)-based probes/composites have been reported for discriminating Cl^−^, Br^−^, and I^−^ to display red/blueshifted PL emissive peaks via anion exchange, as noted in Table 2 [101,102,103,104,105,106,107,108,109,110,111,112]. According to these sensing studies, the presence of anions in the aquatic environment may also lead to a rapid anion exchange due to the disturbed structural parameters. These issues should be addressed with in-depth investigations. Surface stabilization by using suitable capping agents may change lattice features, resulting in an enhanced PLQY; thus, the proposed anion-sensing performance by CsPbX_3_ (X = Cl, Br, and I)-based probes/composites is not yet confirmed and requires further research. The real question is how can the anion-sensing performance be confirmed if the sensing medium itself could affect the stability of the proposed CsPbX_3_ (X = Cl, Br, and I)-based probes/composites. The reaction-based sensing of S^2−^ using tetraphenylporphyrin tetrasulfonic acid (TPPS)-modified CsPbBr_3_ NCs was also observed [113], which showed dependence on composition concentrations. Therefore, the optimization of composition concentrations is regarded as a high-priority task and requires detailed investigations. Due to the instability issues of CsPbX_3_ (X = Cl, Br, and I)-based probes, anion discrimination to distinct competing matrices in real water samples becomes more urgent.

## 5. CsPbX_3_ (X = Cl, Br, and I) and Composites for the Recognition of Chemicals and Explosives

Similar to the quantification of metal ions and anions, CsPbX_3_ (X = Cl, Br, and I) and composites have been widely applied in the discrimination of chemicals and explosives. Yin and co-workers reported the use of CsPbBr_3_ NCs (synthesized with the hot-injection method; PL at 510–520 nm) in the quantification of iodomethane (CH_3_I) via the halide exchange reaction in the presence of oleylamine (OLA) [114]. The presence of OLA induced the nucleophilic substitution of CH_3_I to release iodide species. As illustrated in Figure 11, the iodide exchange occurs rapidly within <5 s, resulting in a redshift of PL (nearly 150 nm; original PL at 660–670 nm). The selectivity of CsPbBr_3_ NCs toward CH_3_I was high with this portable approach. The linear range of CH_3_I detection is between 0.7 and 70 µM with an LOD of 0.2 ± 0.07 µM. Based on the results, this work can be regarded as innovative. Shortly after, Feng et al. described the use of yttrium single-atom-doped cesium lead bromide nanocrystals (Y-SA/CsPbBr_3_ NCs) for detecting CH_3_I [115]. The Y single-atom deposition was carried out using a photo-assisted method. In the presence of OLA, the PL peak of CH_3_I was redshifted due to the halide exchange reaction. The linear range of CH_3_I detection was between 5.6 and 157 µM, with an LOD of 0.3 µM. Except for the anchoring of the Y single atom, this is a follow-up work of an earlier report [114], hence requiring more research efforts to confirm its novelty.

Xie et al. reported a paper-based microfluidic colorimetric assay for dichloromethane/dibromo methane (CH_2_Cl_2_/CH_2_Br_2_) in the presence of trialkyl phosphines (TOP) by using CsPbX_3_ (X = Cl, Br, or I) nanocrystals [116]. When adding TOP or through UV-photon-induced electron transfer, the homogeneous nucleophilic substitution could be enhanced to afford a colorimetric fluorescent response with corresponding peak shifts. In the presence of CH_2_Cl_2_, the CsPbBr_3_ NCs displayed a linear peak shift (from 510 nm to 460 nm) in CH_2_Cl_2_ concentrations of 0–0.9 M, with an LOD of 48 mM. Similarly, In the presence of CH_2_Br_2_, CsPbBr_0.5_I_2.5_ displayed a linear PL peak shift (from 660 nm to 560 nm) in CH_2_Br_2_ concentrations of 7.2–21.0 mM, with an LOD of 1.7 mM. Although the results of this work look appealing, improvement in LODs is required before commercialization. Saikia et al. used cetyltrimethylammonium bromide (CTAB)-passivated CsPbBr_3_ to effectively discriminate ethanol and methanol [117]. Due to the different interaction modes with CTAB, CsPbBr_3_ displayed diverse PL “turn-off/turn-on” responses with LODs down to 9.3 ppb. This technique has been validated in petrol and cough syrup samples; therefore, it is quite innovative.

Bahtiar et al. described the employment of CsPbBr_3_ NCs for quantifying the benzoyl peroxide (PBO) concentration in solutions [118]. When adding oleylammonium iodide (OLAM-I), the PL emission of CsPbBr_3_ NCs at 515 nm was redshifted to 660 nm via the halide exchange reaction. When BPO was added to the above solution, the original PL emission was blueshifted within 1–2 min. Zhang and co-workers also used luminescent CsPbBr_3_ NCs (synthesized with the hot-injection method; PLQY = 87%) to detect BPO [119]. The linear range of BPO detection was observed between 0 µM and 120 µM, with an LOD of 0.13 µM. The applicability of both reports was demonstrated in white flour and noodles. Thus, this is a unique method for BPO detection. Huangfu et al. confirmed the highly responsive photoluminescence sensing performance of CsPbBr_3_ quantum dots (QDs) for total polar material (TPM) identification in edible oils [120]. As seen in Figure 12, the CsPbBr_3_ QDs display diverse colorimetric and PL emissive responses to individual polar solvents (dimethyl sulfoxide (DMSO), dimethyl formamide (DMF), methanol, acetonitrile, ethanol, 1-propanol, acetone, ethyl acetate, chloroform, dichloromethane, and toluene). This TPM detection was effectively applied in edible oils, such as olive oil, soybean oil, and sunflower oil. Hence, it is regarded as an innovative method for the real-time quality assessment of edible oil.

Zhao and co-workers developed orange-emitting oil-soluble CsPbBr_1.5_I_1.5_ QDs for detecting excessive acid number (AN), 3-chloro-1,2-propanediol (3-MCPD), and moisture content (MC) for edible oil quality assessment [121]. The PL emission of the CsPbBr_1.5_I_1.5_ QDs at 609 nm was quenched when detecting excessive acid number (AN). The peak at 609 nm was blueshifted to 583 nm when 3-MCPD was detected. For MC detection, mesoporous silica-coated CsPbBr_1.5_I_1.5_ QDs were adopted as ratiometric sensors to develop water-stable green-emitting CsPbBr_3_ nanosheet (NS) probes. The LODs were determined for the detection of AN, 3-MCPD, and MC as 0.71 mg KOH/g, 39.8 μg/mL, and 0.45%, respectively. Based on these results, this work can be regarded as innovative. Aamir et al. demonstrated the use of CsPbBr_3_ microcrystals for the PL-based detection of nitrophenol [122]. The PL emission was quenched rapidly due to the π–π stacking interaction of the benzene ring with CsPbBr_3_ microcrystals. PL emission was quenched linearly in nitrophenol concentrations of 0.1–0.6 mM. This study reported a preliminary result, thereby requiring more research work.

Chen et al. reported the use of CsPbX_3_ QDs (Br/I; synthesized via the hot-injection method; PLQY = 52.88% and 46.18%, respectively) for the highly selective detection of explosive picric acid (PA) [123]. In which, the green/red fluorescence of CsPbX_3_ (Br/I) at 510 nm/675 nm was quenched in the presence of PA. The linear regression of CsPbX_3_ (Br/I) to PA was in the ranges of 0–180 nM and 0–270 nM, with estimated LODs of 0.8 nM and 1.9 nM, respectively. Based on the supported evidence, the authors speculated that the electrostatic-assisted energy transfer is the possible sensor mechanism. Figure 13 displays a schematic model of the CsPbBr_3_ QD-based quenching response to PA and its paper-strip application. This is outstanding work, but additional research is necessary for commercialization. Aznar-Gadea and co-workers described the consumption of molecularly imprinted CsPbBr_3_ nanocomposites for rapid explosive taggant detection at the gaseous stage [124].

A molecularly imprinted polymer (MIP) sensor was fabricated by embedding CsPbBr_3_ NCs in polycaprolactone (PCL). When exposed to template molecules, such as 3-nitrotoluene (3-NT) and nitromethane (NM), PL quenching responses (>75%) were observed. The MIP sensor showed high selectivity to NT within 5 s, with an LOD of 0.218 mg mL^−1^. This is a preliminary work; hence, it should be further extended for commercialization. Table 3 summarizes the synthetic route, PLQY, linear range, LOD, and application of CsPbX_3_ (X = Cl, Br, and I) and composites toward the detection of chemicals and explosives.

### Critical View on CsPbX_3_ (X = Cl, Br, and I)-Based Chemical and Explosive Sensors

The detection/quantification of specialized chemicals, such as CH_3_I, CH_2_Cl_2_, CH_2_Br_2_, benzoyl peroxide, and excessive acid number (AN), via anion exchange mechanisms [114,115,116,117,118,119,120,121] cannot be regarded as a specific quantification procedure because of its similarity to anion detection. This should be critically investigated to pursue the “state-of-the-art” sensing procedure. Since the observed ratiometric PL responses are also similar to those in the anion sensing studies, critical investigations are required for commercialization. Furthermore, discriminating explosives was demonstrated via the PL quenching response resulting from surface interaction and charge transfer between nitro-containing explosives and CsPbX_3_ (X = Cl, Br, and I)-based probes or composites [122,123,124]. However, this also requires critical studies to justify the exact static/dynamic PL quenching responses.

## 6. CsPbX_3_ (X = Cl, Br, and I) and Composites for the Quantification of Gaseous Analytes and Volatile Organic Compounds (VOCs)

Many CsPbX_3_ (X = Cl, Br, and I) and composites have also been reported for the detection of gaseous analytes and VOCs, as described in this section. Huang et al. reported the oxygen-sensing performance of CsPbBr_3_ NCs [125]. NCs have a porous structure, which allows for the rapid diffusion of O_2_, resulting in PL quenching. The underlying sensor mechanism is that O_2_ molecules are directly involved in the extraction of photogenerated electrons from the conduction band of CsPbBr_3_ NCs. This work lacks in-depth sensor investigations, thereby requiring extensive research. Lin et al. developed Mn^2+^-doped cesium lead chloride nanocrystals (Mn:CsPbCl_3_ NCs) by using a heat-up strategy for sensing O_2_ via luminescent dopants and the host–dopant energy transfer mechanism [126]. As seen in Figure 14a–e, upon exposure to O_2_, the phosphorescence intensity of Mn:CsPbCl_3_ NCs decreased linearly between 0 and 12% of O_2_. High sensing reversibility, rapid signal response, and high photostability in air were also demonstrated. The estimated Stern–Volmer quenching constant (K_SV_) value following first-order kinetics was 0.0658% [O_2_]^−1^ (R^2^ = 0.9997). This work is innovative, but it lacks practical applications.

Brintakis and co-workers described the use of CsPbBr_3_ nanocubes as self-powered ozone sensors, which showed higher sensitivity (54% in 187 ppb) and faster responses (between 100 s and 150 s) and recovery (between 250 s and 320 s) [127]. The sensor response of CsPbBr_3_ nanocubes to O_3_ was recorded between 4 and 2650 ppb (ppb = parts per billion). The as-synthesized CsPbBr_3_ nanocubes were semiconductors with a certain resistance. When exposed to O_3_, an accumulation layer of holes with lower resistance covering the whole surface of the nanocubes was formed resulting in an increase in electrical current. This is an innovative work, which has attracted many scientists to work in this direction. Park et al. fabricated cesium lead bromide nanofibers (CsPbBr_3_ NFs) by attaching CsPbBr_3_ NCs with cellulose nanofibers (CNFs) for N_2_-sensing investigations [128]. When exposed to N_2_ flow, the PL intensity of the CsPbBr_3_ NFs at 520 nm was quenched linearly between 1 and 20 ppm (R_2_ = 0.99433; ppm = parts per million) with an LOD of 1 ppm. The surface trapping of N_2_ was proposed as the underlying mechanism for N_2_ detection. This is a study on a conventional sensor, but further research is still necessary.

Nanocrystalline ZnO sensitized with CsPbBr_3_ NCs for the photoresistive sensing of NO_2_ gas was proposed by Chizhov and co-workers [129]. In a temperature range of 25–100 °C, the ZnO/CsPbBr_3_ nanocomposite showed a linear sensor response between 0.5 and 3.0 ppm to NO_2_. These sensing measurements were conducted under periodic blue LED illumination (light = t_dark_ = 20 s; LED = light emitting diode). At 1 ppm of NO_2_ gas, it was found that the optimum temperature to provide the best reversibility of sensor measurements was 75 °C. Under periodic illumination, the increase in the electron concentration of ZnO led to the adsorption of oxidizing molecules over the surface. The photoexcited holes in CsPbBr_3_ were involved in the redox reaction, resulting in a change in the electrical signal. This research is an impressive report, and it could be further developed for commercialization. Yueyue et al. demonstrated the use of CsPbBr_3_ QD/ZnO MB nanocomposites (MBs = microballs) for NO_2_ detection at room temperature [130]. In the presence of diverse CsPbBr_3_ QDs (0.5 wt%, 1.0 wt%, and 1.5 wt%) with ZnO MBs, the sensing responses to NO_2_ were recorded. ZnO MB composited with 1.0 wt% QDs displayed a greater photoresistive response to NO_2_ (for 5 ppm NO_2_; R_gas_/R_air_ = 53; response/recovery time = 63s/40s under 520 LED (1.2 W/m^2^) illumination). The adsorption and desorption of NO_2_ over the ZnO surface were proposed as the underlying mechanism, and the QDs had minimal effect on the sensor mechanism. This is a well-developed study on NO_2_-sensing studies of CsPbBr_3_-ZnO conjugates, but more studies on the interference and real-time applications are required.

The ultrafast sensing of NO_2_ was demonstrated using FA_0.83_Cs_0.17_PbI_3_ (FAC) prepared via one-step spin-coating in ambient conditions [131]. When exposed to NO_2_ (20, 10, 5, 2, 1, and 0.5 ppm), the photoresistive responses (R_gas_/R_air_) were measured as 2.64, 2, 1.52, 1.27, 1.17, 1.1, respectively. The optimum sensor response of NO_2_ was recorded at 10 ppm, with a response/recovery time of 2 s/22 s. The strong oxidizing ability of NO_2_ gas attracts electrons from FAC, which is a p-type semiconductor. The calculated values of the adsorption energy (E_ads_) of NO_2_ of FA^+^ and Cs^+^ in FACs are −0.37 and −0.60 eV, respectively, thereby allowing for the spontaneous adsorption of NO_2_. It should be noted that NO_2_ molecules attached to FACs may induce lattice distortion, which leads to a dipole moment and the migration of charge carriers. All these factors were involved in the sensing of NO_2_. The roles of FA^+^ and Cs^+^ in detecting NO_2_ were clearly justified. Further research in this direction is required to improve the sensitivity. Wang and co-workers combined ZnO nanorods with CsPbBr_3_/Cs_4_PbBr_6_ particles to afford the CsPbBr_3_/Cs_4_PbBr_6_/ZnO composite, which showed photoresistive sensor responses to NO (100 ppm), with the R_gas_/R_air_ reaching 2296 (at 50 °C) and an LOD of 1 ppm [132]. The response and recovery time were determined as 1235 s and 173 s, respectively. NO gas attracts electrons from the surface of CsPbBr_3_/Cs_4_PbBr_6_/ZnO to release O_2_, which results in electrical signal changes. Although electrons were accumulated in the conduction band of ZnO, the diffusion of NO gas was hindered due to the covering layer of CsPbBr_3_/Cs_4_PbBr_6_ particles. Therefore, a long response time was recorded. This work requires further research for the optimization of the response/recovery time and interference studies.

Chen et al. described the use of luminescent CsPbBr_3_ QDs (synthesized via the sonication method) for the solution-mediated detection of hydrogen sulfide (H_2_S) [133]. As seen in Figure 15A,B, the luminescent intensity of QDs at 520 nm was quenched linearly between 0 and 100 µM with an LOD of 0.18 µM. Note that the QDs exhibit a greater selectivity to H_2_S among all the interfering species, as shown in Figure 15C. H_2_S penetrated the surface of QDs and reacted with Pb^2+^ to form PbS, which resulted in fluorescent quenching. The applicability of this work was demonstrated in rat brain samples; therefore, it can be considered a unique report. Luo and co-workers synthesized a CsPbBr_3_@CMO nanocomposite by encapsulating CsPbBr_3_ QDs with cetyltrimethylammonium bromide and mineral oil via sonication. The composite was stable in water and was applied for detecting H_2_S [134]. When adding H_2_S, the PL emission of CsPbBr_3_@CMO at 524 nm was quenched linearly in the range of 0.15–105 µM, with an estimated LOD of 53 nM. The sensor response was attributed to the formation of PbS originating from H_2_S and excessive Pb^2+^ present in CsPbBr_3_ QDs. The high selectivity and applicability of CsPbBr_3_@CMO to H_2_S were confirmed by interference studies and rat brain investigations.

A water-soluble CsPbBr_3_@sulfobutylether-β-cyclodextrins nanocomposite was synthesized via sonication and employed as a photothermal sensor [135]. The H_2_S acted as a switch to trigger a photothermal response, which resulted in PL quenching (at 520 nm). The linear regression of H_2_S was observed between 0.5 µM and 6 mM with an LOD of 0.3 µM. This work involved zebra fish-based in vivo studies, but it lacks interference investigations; therefore, further research is mandatory. Shan et al. reported the utilization of tributyltin oxide (TBTO)-capped CsPbBr_3_ QDs (CsPbBr_3_-Sn QDs) for the chemoresistive sensing of H_2_S [136]. The sensor response (R_gas_/R_air_) at 100 ppm H_2_S reached 6.69, with a response/recovery time of 278 s/730 s. The sensitivity of H_2_S could reach 0.58 at 250 ppb. During the adsorption of H_2_S, the charge distribution of the internal CsPbBr_3_-Sn QDs was affected by a metalloorganic TBTO molecule, thereby leading to enhanced sensing performance. On the other hand, the sensor performance was affected by the interaction between H_2_S and CsPbBr_3_ QDs through PbS formation. This is an innovative work, but the response still needs to be improved with more interference studies.

Chen et al. described the fabrication of photoresistive sensors comprising a porous network of CsPbBr_3_, which can generate an open-circuit voltage of 0.87 V under visible light irradiation, to be employed in O_2_ and VOC (acetone and ethanol) detection [137]. The device showed 100% photocurrent enhancement for O_2_ with a corresponding response and recovery time of 17 s and 128 s under visible light irradiation. At 1 ppm of acetone/ethanol (at 30 °C; illumination density = 37.8 mW cm^−2^), the device displayed sensor responses (I_VOC_/I_air_ − 1) of 0.03 and 0.025 with a response/recovery time of >200 s/400 s, respectively. The sensor response was attributed to surface lattice changes. This work needs further optimization to improve the sensor response and response/recovery time before commercialization. Xuan et al. proposed the use of stable ZnO-coated CsPbBr_3_ NCs (CsPbBr_3_@ZnO NCs, synthesized using an in situ technique) for the photoresistive detection of heptanal (breath biomarker) at room temperature [138]. The sensor response of CsPbBr_3_@ZnO NCs at 200 ppm heptanal was measured as S = 0.36 (S = I_h_ − I_0_/I_0_, where I_h_ and I_0_ represent the current values in the presence and absence of heptanal gas, respectively) with response/recovery time of 36.5 s/5.3 s. Note that the LODs were down to 2 ppm in air and 3 ppm under artificial conditions. The heptanal-induced lattice distortion was attributed as the underlying mechanism for the sensor response. This method can facilitate the early detection of lung cancer and COVID-19. This is an innovative work and should be extensively studied with interfering species toward biomedical applications. CsPbBr_2_I was also reported as a self-powered sensor for reducing and oxidizing gas molecules via surface adsorption and desorption [139]. Because this device detects multiple gaseous analytes, the possibility of an interfering effect cannot be ruled out.

Liu and co-workers reported triethylamine (TEA) detection by using CsPbBr_3_-decorated ZnO polyhedrons derived from ZIF-8 [140]. ZnO-CsPbBr_3_ showed a higher photoresistive sensor response to TEA (~60 for 100 ppm at 180 °C) than pristine ZnO and ZnO NP-CsPbBr_3_. Note that the ZnO-CsPbBr_3_ also displayed shorter response and recovery times of 2 s and 18 s with an LOD of 5 ppb. The sensor response is due to the adsorption of oxygen molecules in the air onto the surface of ZnO-CsPbBr_3_, which generates oxygen anions to initiate a redox reaction when exposed to TEA. Based on the reported sensing performance, this work is innovative, but it requires more research to optimize interference studies. Xu et al. demonstrated the sensing performance of CsPbBr_3_ to ethanolamine (EA), in which a high response (R_gas_/R_air_ = 29.87 for 100 ppm EA) with a response/recovery time of 62 s/782 s and an LOD of 21ppb were reported [141]. Figure 16 shows the EA sensing performance of CsPbBr_3_, reversibility, and linear ranges to EA. Following the reaction Formulas (2) and (3), the adsorbed O_2_ molecules on the CsPbBr_3_ surface generate O_2_^−^ anions, which further reduce EA to generate sensor signals.
O_2_(gas) + e^−^ → O_2_^−^ (ads)(2)
H_2_NCH_2_CH_2_OH + 3O_2_^−^ → NH_2_OH + 2CO_2_ + 2H_2_O + 3e^−^(3)

Shortly after, the same research group as in Ref. [141] demonstrated using the 3-mercaptopropionic acid (MPA)-regulated heterojunction of CsPbBr_3_ NPs/ZnO NPs for detecting EA [142]. The hydrophilic groups in MPA enhance the stable anchoring of ZnO over the CsPbBr_3_ surface via hydrogen bond-facilitated MPA network structures. The O_2_ molecules anchored on ZnO generated O_2_^−^ species, which interacted with EA via a redox reaction to generate a sensor signal. At 100 ppm of EA, CsPbBr_3_-2MPA/ZnO displayed a chemoresistive sensor response of 13.23 with a response/recovery time of 50 s/698 s and an LOD of 31 ppb. This work is innovative, judging from its supportive evidence in interference studies and the justification of the underlying mechanism. Nevertheless, further optimization to maximize the sensor signal is still required. CsPbBr_3_ NC-anchored amine-functionalized graphene oxide (GO) was demonstrated in the electrochemiluminescence (ECL)-based detection of cupric oleate in acetonitrile containing 10 mM of tripropylamine (TPrA) [143]. The ECL response for the cupric oleate showed a decreasing trend in the range of 10^−18^−10^−16^ M with LODs down to the attomolar (10^−18^ M) level. This is a preliminary study, thereby requiring additional efforts.

Thiophene sulfides are one of the harmful contaminants of air pollutants; hence, their detection becomes vital. Feng and co-workers proposed the use of CsPbBr_3_ NCs and CsPbBr_3_/SiO_2_ NCs for detecting diverse thiophene sulfides [144]. Benzothiophene (BT), dibenzothiophene (DBT), 2-methylbenzothiophene (2-MeBT), 3-methylthiophene (3-MeBT), and thiophene (TP) were discriminated using the fluorescent quenching method, as seen in Figure 17. The linear regression of BT, DBT, 2-MeBT, and TP detection was observed between 10 and 50 ppm. As for t3-MTP, linearity was observed between 20 and 50 ppm. The fluorescence of perovskite NCs can be effectively weakened by thiophene sulfides to varying degrees due to the different interactions between thiophene sulfides and CsPbBr_3_ NCs and CsPbBr_3_/SiO_2_ NCs. Hence, this method can be adopted for both quantitative and qualitative detection of thiophene sulfides. This work is impressive in terms of its qualitative and quantitative measurements.

A method involving dynamic passivation over the surface of CsPbBr_3_ QDs (synthesized using the hot-injection method) was proposed by Huang et al. for the PL-enhanced detection of ammonia (NH_3_) [145]. In this study, the luminescence of purified QD film was enhanced when exposed to NH_3_. The linear range of PL enhancement at 610 nm was between 25 and 300 ppm with an LOD of 8.85 ppm. The photoluminescent response and recovery times were determined as 10 s and 30 s, respectively, at room temperature. In particular, this innovative work explains the analysis at room temperature. Following a similar approach, the employment of CsPbBr_1.5_Cl_1.5_ QDs, CsPbBr_3_ QDs, and CsPbBr_1.5_I_1.5_ QD films for the PL “turn-on” detection of ammonia (NH_3_) was demonstrated [146]. As illustrated in Figure 18, the QD films display exceptional selectivity among other interferences. All these films display good linear behavior (25–200 ppm) and LODs of ≈20 ppm. The uniqueness of this passivation method is well demonstrated by these reports, and hence it can be extended for developing commercialized devices for the detection of ammonia (NH_3_). Similar to the “turn-on” detection, a few CsPbX_3_-conjugated materials were also proposed for the discrimination and quantification of NH_3_ via PL quenching responses. Humidity-resistant CsPbBr_3_–SiO_2_ nanocomposites, porous nanofibers/nanocomposites (CsPbBr_3_ NFs and CsPbBr_3_/BNNF; BNNF = boron nitride nanofiber), and stable CsPbBr_3_ QDs grown within Fe-doped zeolite X were proposed for the PL-quenched detection of NH_3_ [147,148,149,150]. The adsorption and desorption of NH_3_ over the surface of these nanocomposites resulted in reversible cycles with given linear ranges and LODs. Table 4 summarizes the synthetic route, PLQY, linear ranges, LODs, and applications of CsPbX_3_ (X = Cl, Br, and I) and composites toward the detection of gas and VOCs.

### Critical View on the Detection of CsPbX_3_ (X = Cl, Br, and I)-Based Gases and VOCs 

Combining CsPbX_3_ (X = Cl, Br, and I) with different materials can result in composited materials with exceptional electro-optical properties and less defect, which can be adopted in the design of PL-based probes, electrochemiluminescence probes, chemoresistive sensors for discriminating N_2_, O_2_, H_2_S, tripropylamine (TPrA), thiophene sulfides, and NH_3_ [125,126,127,128,129,130,131,132,133,134,135,136,137,138,139,140,141,142,143,144,145,146,147,148,149,150]. To achieve the above goal, compositing ratios need to be optimized. Changes in the compositing ratios may significantly affect selectivity, thereby requiring careful/critical adjustments. The detection of H_2_S was demonstrated in rat brain and zebrafish studies [133,134,135], but there is no clear indication of how to overcome the toxicity induced by Pb^2+^ in CsPbX_3_. The film- or test-strip-based sensing of NH_3_ mostly displayed dependence on the crystalline and morphological features of CsPbX_3_ (X = Cl, Br, and I) and composites. Thus, optimizing the crystallinity/morphology of thin film is critical to attaining the best results.

## 7. Humidity, Temperature, and Radiation/Photodetection by CsPbX_3_ (X = Cl, Br, and I) and Composites

Humidity and moisture are the main causes resulting in perovskite material degradation, which affects the practical use and commercialization of perovskite-based energy devices [151]. The water molecules in the air react with the metal halide perovskite surface, which rapidly affects the morphology and uniformity, resulting in changes in optical properties and conductivity [152]. The above effect is the major mechanism of humidity-sensing responses. Doping with specified metal ions can improve the environmental stability of metal halide perovskites and reduce the humidity effect on the surface/morphology [153]. Due to the structural distortion (phase change) and instability of CsPbBr_3_, its composites can be effectively employed for the trace detection of water and humidity (%RH). For example, dimethyl aminoterephthalate-functionalized CsPbBr_3_ QDs (CsPbBr_3_@DMT-NH_2_ QDs) were synthesized with a low-temperature method and engaged in trace water detection in edible oils [154]. The PL emission at 530 nm was quenched in trace water (0.05–5%; *v*/*v*) with ratiometric enhancement at 445 nm. The LOD (3σ/slope) and limit of quantification (LOQ; 10σ/slope) were 0.01% and 0.04%, respectively. The involvement of the IFE, FRET, aggregation of QDs, and disintegration were proposed as the underlying mechanisms. This work is notable and can be extended to detect water traces in oils and chemicals. Thereafter, Xiang and co-workers described the luminescent quenching response of CsPbBr_3_ to trace water in herbal medicines [155]. The PL intensity at 503 nm was quenched linearly with water contents of 1–17%. The values of LODs in Seutellaria baicalensis and Astragalus flavone were 0.75% and 0.67%, respectively.

These materials also show a great response to varying humid conditions with a linear behavior between 33 and 98% RHs and an estimated LOD of 12% RH, as visualized in Figure 19. A phase change from CsPbBr_3_ to CsPb_2_Br_5_ was attributed as the mechanism for the observed PL quenching response to water and RH. The recovery of RH in the above herbals was in the range of 96.7–102.5%; therefore, this is noted as an inspiring work. The utilization of CsPbBr_3_ NPs in impedance-based humidity-sensing (under 20 mV) studies was also proposed [156]. The CsPbBr_3_ NPs were operative in the humidity range of 11–95% with a response/recovery time of 2.8 s/9.7 s and a sensitivity of 1.56% RH. This is also an inspiring work that can be further explored in commercial electrochemical device fabrication.

Variations in temperature led to changes in phase and grain sizes [157], which can be adopted in temperature sensors via monitoring changes in I–V responses, PL intensity, absorbance, etc. On the other hand, alterations in temperature also affected the grain uniformity and morphologies of CsPbX_3_ and composites [158], resulting in changes in the PL intensity and current density. However, the doping of metal ions may also improve the temperature sensitivity of CsPbX_3_ and composites [159], as illustrated in this section. The doping of metal ions may enhance the sensitivity of CsPbX_3_ and composites. For example, Chang et al. described the temperature-sensing ability of Mn^2+^-doped CsPbCl_3_ QDs (CsPbCl_3_:0.1Mn^2+^ QDs; PLQY = 47.3%) via a dual-mode luminescent response at 298–353 K [160]. In this study, the stability of QDs was improved by replacing Pb^2+^ with Mn^2+^. For the 6% Mn^2+^-doped CsPbCl_3_ QDs, the PL intensity at 410 nm and 600 nm was quenched considerably compared with those of undoped CsPbCl_3_ QDs (quenching was observed at 410 nm only), as seen in Figure 20.

The 6% Mn^2+^-doped CsPbCl_3_:QDs displayed 83% and 22% quenching at 410 nm and 600 nm, respectively. Using the fluorescence intensity ratio (FIR) and full width at half-maximum (FWHM), the maximum relative sensitivity (SR) of CsPbCl_3_:0% Mn^2+^ was 7.38% K^−1^ at 298 K and 2.13% K^−1^ at 353 K. Although this is a follow-up work of earlier reports [161,162], which also demonstrated the temperature-sensing ability through the dual-mode fluorescent response, its results are impressive. Moreover, Mn^2+^-doped CsPbCl_3_@glass [161] and Mn^2+^-doped CsPbCl_3_ NCs [162] were employed as dual-mode luminescent sensors in temperature ranges of 80–293 K and 80–30 K, respectively.

Similar to Mn^2+^ doping, Eu^3+^-doped cesium lead halide perovskite glasses (Eu^3+^:CsPbCl_2_Br_1_ QDs and Eu^3+^:CsPbBr_3_ QDs) were also used as effective materials in optical temperature-sensing studies [163,164]. With increasing temperature from 80 K to 440 K and 93 K to 383 K, both Eu^3+^:CsPbCl_2_Br_1_ QDs and Eu^3+^:CsPbBr_3_ QD glasses displayed blueshifted PL quenching at 458 nm (excitation at 395 nm) and 519 nm (excitation at 394 nm), respectively. For Eu^3+^:CsPbCl_2_Br_1_ QD glass, the absolute temperature sensitivity maxima (S_a_) and the relative temperature sensitivity (S_r_) were established as 0.0315 K^−1^ and 3.097%/K, respectively. Similarly, the S_a_ and S_r_ values of Eu^3+^:CsPbBr_3_ QD oxyhalide glass were 0.0224 K^−1^ and 2.25% K^−1^. Figure 21 shows the optical quenching response of one of the Eu^3+^:CsPbBr_3_ QD oxyhalide glass samples (S3) between 93 K and 383 K. These results also suggest that metal doping can enhance the temperature-sensing ability of cesium lead halide perovskites and composites for future device development.

CsPbCl_x_Br_3−x_ NCs confined in hollow mesoporous silica (CsPbCl_1.2_Br_1.8_ NCs@h-SiO_2_) and integrated with K_2_SiF_6_:Mn^4+^ phosphor in the EVA polymer matrix (EVA = ethylene–vinyl acetate) were proposed by Huang and co-workers for reversible temperature sensing [165]. The high density of halide vacancies played a vital role in temperature sensing between 30.0 °C and 45.0 °C. The temperature sensitivity was established as 13.44% °C^−1^ at 37.0 °C. This is an inspiring work that investigates defects in sensor performance. Similarly, microencapsulated CsPbBr_3_ NCs with K_2_SiF_6_:Mn^4+^ phosphor (CsPbBr_3_-KSF-PS film) showed temperature sensitivity between 30 °C and 70 °C, with the S_r_ value reaching 0.31% °C^−1^ at 45 °C [166]. A highly stable film with its fluorescence visible to the naked eye (green-to-red fluorescence) was achieved by optimizing the CsPbBr_3_ and KSF ratio and applied for temperature sensing. Lu et al. developed dual-phase compounds containing CsPbBr_3_ QDs (emission at 529 nm) and NaYF_4_:Ho^3+^ NPs (emission at 647 nm and 751 nm) for conducting optical temperature sensing between 293 and 433 K at an excitation wavelength of 447 nm [167]. The nonradiative recombination between the energy band and the defective surface led to the thermal quenching of CsPbBr_3_. S_a_ and S_r_ values of 385 K^−1^ and 5.13% K^−1^ were reported in this work. This is an interesting report among all the studies on temperature sensors.

The use of cesium lead halides and composites in radiation/photo detection was proposed by numerous researchers [168,169,170,171,172,173,174,175,176,177,178,179,180]. Ti/Ni/CsPbBr_3_/Ni/Ti (with ER of 5.70% at 8250 V·cm^−1^; ER = energy resolution) [153], single crystals of CsPbBr_3_ (with diverse hole mobilities) [169,170,171,172,175], In/LiF/CPB/Au detectors [173], CsPbBr_3_ QDs embedded in P_3_HT:PC_61_BM (P_3_HT:PC_61_BM:QD; a bulk heterojunction photodiode) [174], CsPbBr_3_ microcrystals on ITO functional substrate [176], a single-crystalline thin film of CsPbBr_3_ (CsPbBr_3_ SCF; switching ratio = 3.2 × 10^3^, and response time = 200/300 ns) [177], CsPbBr_3_/RGO nanocomposites (RGO = reduced graphene oxide) [178], CsPbBr_3_ QD/ZnO NWs nanocomposites [179], and ZnONW/CsPbBr_3_ QD/graphene heterojunction were demonstrated for X-ray, radiation, and photodetection. Ion migration, photon–exciton coupling, electron–hole diffusion, and strain-based mechanisms were proposed as the underlying mechanisms for the radiation/photodetection. Most of the proposed cesium lead halide-based composites showed exceptional performance. For example, P_3_HT:PC_61_BM:QD [174] displayed good current density and X-ray photocurrent response by optimizing the weight ratio, as depicted in Figure 22. Since there are several reviews available on the radiation/photodetection performance of perovskites [14,15,16,17,18,19,32,61,63], they will not be discussed here.

### Critical View on the Detection of CsPbX_3_ (X = Cl, Br, and I)-Based Humidity, Temperature, and Radiation/Photodetection

The low stability and degradation of CsPbX_3_ (X = Cl, Br, and I) and composites were adopted as sensor responses for humidity detection [151,152,153,154,155,156]. However, reports on = humidity sensors also discussed the recovery, which is problematic due to the environmental instability issue of the CsPbX_3_ (X = Cl, Br, and I) and composites. Thus, CsPbX_3_ (X = Cl, Br, and I) and composite-based recovery of humidity sensors must be carefully examined. Phase transitions may occur in CsPbX_3_ (X = Cl, Br, and I) and composites during temperature sensing [157,158,159,160,161,162,163,164,165,166,167]; thus, in-depth investigations are critical in many cases involving phase transitions. Likewise, doping of ions, such as Mn^2+^, may enhance the phase changes, which requires a more careful examination. When detecting photon/radiation, lattice defects could be generated in CsPbX_3_ (X = Cl, Br, and I). This problem can be alleviated by using composites with diverse materials [168,169,170,171,172,173,174,175,176,177,178,179,180]. However, the compositing ratios must be critically evaluated to achieve better sensing results.

## 8. CsPbX_3_ (X = Cl, Br, and I) and Composites in the Detection of Bioanalytes, Drugs, Fungicides, and Pesticides

Cesium lead halides and composites were applied in discriminating biologically significant analytes, as described in this section. Niu and co-workers constructed a dual-emitting nanoprobe consisting of CsPbBr_3_ NCs and red-emissive Cu NCs (CsPbBr_3_@Cu nanohybrid) for the luminescent detection of hydrogen peroxide (H_2_O_2_) and glucose [181]. CsPbBr_3_@Cu nanohybrid with PL emission centered at 517 nm and 645 nm displayed ratiometric responses to H_2_O_2_ and glucose. A linear ratiometric response at F_645_/F_517_ was observed in the ranges of 0.2–100 µM and 2.0–170.0 μM with LODs of 0.07 µM and 0.8 μM, respectively, when recognizing H_2_O_2_ and glucose. The electron-transfer-induced redox reaction was considered to be the underlying sensor mechanism. This method was also demonstrated for glucose detection in human serum samples with an impressive RSD value of < 4%. The TiO_2_/CsPbBr_1.5_I_1.5_ composite film and CsPbCl_3_/TiO_2_ served as inverse opal electrodes for the photoelectrochemical discrimination of dopamine (DA) and alpha-fetoprotein (AFP), respectively [182,183]. Slow volatilization was used to fabricate electrodes. The detection of DA showed a linear response between 0.1 and 250 μM, with an LOD of 12 nM, and the detection was also demonstrated in human serum samples [167]. Therefore, it is noted as an impressive report. On the contrary, the detection of AFP showed a linear regression between 0.08 and 980 ng/mL with an LOD of 30 pg/mL [183]. However, this report lacks information on real-time applications.

Saikia et al. reported the utilization of CsPbBr_3_ microcrystals (via one-pot synthesis; PLQY = 60%) as a sensing probe for the fluorometric detection of uric acid (UA) via hydrogen bonding interactions [184]. The PL of CsPbBr_3_ microcrystals at 520 nm (green to blue) was dynamically quenched within a response time of 30 s in the presence of UA. The linear regression of UA was observed between UA concentrations of 3.1 nM and 1.33 µM with an LOD of 0.063 ppm. This is a remarkable work with applications in human serum samples. Wang and co-workers described a ratiometric fluorescent approach for sensing acetylcholinesterase (AChE; 57 kDa protein; dispenses nerve impulse spread) by using the CsPbBr_3_ NC-TPPS nanocomposite (synthesized via self-assembly strategy; PLQY = 60%; TPPS = Tetraphenylporphyrin tetrasulfonic acid) [185]. The ratiometric response at F_520_/F_650_ to AChE showed a linear response between 0.05 and 1.0 U/L, with an LOD of 0.0042 U/L. Investigations of AChE quantification in human serum samples displayed >95% recovery, with an RSD of <4%. This is an interesting work with low interfering effects and real-time applicability. CsPbBr_3_ QDs and CsPbBr_3_ QD/MoS_2_ (MoS_2_ = molybdenum sulfide) nanoflakes were proposed as chemiluminescence biosensors for human hepatitis B, immunodeficiency virus, and AFP via the sandwich complex formation [186]. In this work, the CsPbBr_3_ QD/MoS_2_ nanoflakes were fabricated by including a parylene-C passivation layer. This is an interesting photosensor for multianalyte detection, which can be extended for commercialization.

Hu et al. reported the use of water-stable CsPbX_3_ (X = Br/I) as a probe for the sensitive detection of penicillamine (PA) [187]. Firstly, CsPbBr_3_ (PL maxima at 525 nm) interacted with iodide via an anion exchange reaction to afford CsPbX_3_ (X = Br/I), which displayed PL emission at 580 nm. As shown in Figure 23, the PL intensity increases accompanied by a blueshift when adding PA between 5.0 and 35.0 nM. The linear regression of PA quantification was observed between 5.0 and 35.0 nM, with LOD values of 1.19 nM (PL intensity vs. PA concentration) and 5.47 nM (PL peak shift vs. PA concentration). Based on the results from high-angle annular dark-field (HAADF) imaging and multiple investigations with organic S-containing substances, the interaction of the sulfhydryl group present in penicillamine with iodide is identified as the underlying sensor mechanism. This is an inspiring work in detecting PA in water. Through a one-pot synthetic method, stable carboxyl group-functionalized CsPbBr_3_–COOH QDs were developed by using amino-poly(ethylene glycol)-carboxyl and perfluorooctyltriethoxylsilane as ligands and were engaged in the PL “turn-on” detection of Mycobacterium tuberculosis (Mtb) [188].

CsPbBr_3_–COOH QDs were composited with MoS_2_ and deoxy nucleic acid (DNA) to afford CsPbBr_3_ QD-DNA/MoS_2_ for the effective detection of Mtb via PL enhancement. Mtb showed a linear response between 0.2 and 4.0 nM, with an LOD of 51.9 pM. This work involved the clinical analysis of tuberculosis pathogens and thereby is noted as an innovative work. CsPbBr_3_ QDs composited with MoS_2_ and passivated with parylene (C, N, and F) were utilized in microbial detection [189] and photo-sensing of bioanalytes. An anodic electrochemiluminescence-based assay of alkaline phosphatase (LOD = 0.714 mU L^−1^; mU = milli units) was also proposed using CsPbBr_3_ QDs [190]. In both reports, charge/electron transport plays a vital role in the detection mechanism.

Li and co-workers used phospholipid-coated CsPbBr_3_ NCs for the effective detection of pore-forming biotoxins and prostate-specific antigens via dual-readout assays [191,192]. By using fluorometric and electrochemical assays, the pore-forming biotoxins were detected by CsPbBr_3_ NCs@PL, which showed a linear regression between 50 nM and 150 µM with an LOD of 50 nM [160]. Likewise, the detection of prostate-specific antigens by CsPbBr_3_ NCs@PL displayed a linear PL enhancement and colorimetric response in the ranges of 0.01–80 ng/mL and 0.1–15 ng/mL, with calculated LODs of 0.081 ng/mL and 0.29 ng/mL, respectively [192]. Both reports were applied in bacterial and clinical analyses, and they can be noted as innovative bioanalytical research. Qi et al. designed a composite consisting of of aptamer-functionalized CsPbBr_3_ NCs and magnetic nanoparticles of Fe_3_O_4_ (MNPs), namely the “Apt-PNCs@cDNA-MNPs” material, for detecting peanut allergen Ara h1 in food samples [193]. In their study, the CsPbBr_3_ NCs were employed as PL labeling probes for collecting fluorescent data. Also, the interaction of Ara h1 with aptamer resulted in PL recovery. The Apt-PNCs@cDNA-MNPs showed linear PL enhancement between 0.1 and 100 ng/mL, with an estimated LOD of 0.04 ng/mL. In terms of the reported real-time studies in food samples, this research can be regarded as inspiring work and should be extended toward food safety monitoring.

Similar to bioanalytes, cesium lead halide perovskites were also employed in the selective and sensitive quantification of drugs. 3-Aminopropyltriethoxysilane-functionalized CsPbBr_3_ QDs (APTES-IPQDs), mesoporous silica nanoparticle-composited CsPbBr_3_ QDs (LMSNs@IPQDs), perofluorooctyltriethyloxylsilane fluorocarbon-assembled Cs_4_PbBr_6_/CsPbBr_3_ NPs (CPB-PFOS), molecularly imprinted CsPbBr_3_ QDs (IPQDs@MIPs), and CsPbBr_3_ QD/BN composites (BN = boron nitride) were used in detecting tetracycline (TC; antibiotic drug) by means of photoinduced electron transfer (PET) or the inner-filter effect (IFE) mechanism [194,195,196,197,198]. All these materials displayed linear TC detection ranges in micro- to millimolar levels with estimated LODs at a nanomolar or micromolar concentration, as listed in Table 5. Moreover, these reports can further attest to the applicability of TC detection in food, water, and soil samples for possible commercialization. Figure 24 illustrates the PL quenching response of CPB-PFOS to TC and the related linear response [196].

Shi et al. described the sensing utility of CsPbBr_3_ NCs toward the quantification of ciprofloxacin hydrochloride (an antibiotic) via PL peak shifting induced by anion exchange [199]. In the presence of ciprofloxacin hydrochloride, the transformation from CsPbBr_3_ NCs to CsPbBr_(3−x)_Cl_x_ NCs occurred together with a corresponding peak shift from 513 nm to 442 nm via anion exchange. The PL peak shift was achieved between 0.8 and 50 mM, with an LOD of 0.1 mM. Though this work was applied in colorimetric paper-strip analysis, additional research is still required to lower the LODs. Through an in situ hot-injection method, CsPbBr_3_-loaded MIP nanogels were developed and employed for discriminating roxithromycin (ROX; an antibiotic) via PL quenching responses [200]. The nanogels showed linear PL quenching behavior between 100 pM and 100 nM, with an LOD of 20.6 pM in detecting ROX. Phase transformation and structural decomposition were proposed as the mechanisms underlying ROX detection. This work was applied in animal-derived food analysis, and hence it can be regarded as an inspiring work toward biomedical applications. Salari and co-workers adopted CsPbBr_3_ QDs (PLQY = 42%) in an organic phase, together with Fe(II) and K_2_S_2_O_8_, in an aqueous medium for the chemiluminescence-based detection of cefazolin (CFZ; an antibiotic) [201]. A chemiluminescence response of CsPbBr_3_ QDs in the presence of Fe(II) and K_2_S_2_O_8_ was observed in CFZ concentrations of 25–300 nM with an LOD of 9.6 nM at pH 7. This report was demonstrated with recoveries of >90% in multiple real samples, such as human plasma, urine, water, and milk samples. Although this is an interesting work, further research must be conducted to optimize experimental conditions.

Water-stable luminescent CsPbBr_3_/Cs_4_PbBr_6_ NCs were synthesized using a water emulsion technique and demonstrated for sensing folic acid (FA; vitamin B) [202]. The CsPbBr_3_/Cs_4_PbBr_6_ NCs showed a linear PL quenching response between 10 and 800 µM of FA, with an LOD of 1.695 µM. The quenching response was attributed to the electrostatic mechanism between NCs and FA. This work was demonstrated through a urine sample-based recovery (>99% with <0.5% RSD). It is a preliminary work on cesium lead halide-based FA detection. He et al. fabricated molecularly imprinted polymer-encoded CsPbX_3_ (X = Cl, Br, and I) microspheres for quantitatively detecting Sudan I (a food colorimetric enhancer) [203]. The PL emission of CsPbBr_3_, CsPbCl_1.5_Br_1.5_, and CsPbI_2_Br microspheres was recorded at 463 nm, 508 nm, and 644 nm, respectively. Interestingly, MIP-CsPbBr_3_ microspheres showed a greater response to Sudan I than nonimprinted ones (NIP-CsPbBr_3_), as shown in Figure 25. A linear range of between 2 and 604 nM with an estimated LOD of 1.21 nM was recorded for detecting Sudan I by MIP-CsPbBr_3_. This work was effectively applied in foodstuffs (egg and chili), which showed >95% recovery. Therefore, it can be utilized in commercial food safety monitoring. Thereafter, barium sulfate-coated cesium lead bromide nanocrystals (CsPbBr_3_ NCs@BaSO_4_) were proposed for the PL-enhanced quantitation of melamine [204]. The PL emission of CsPbBr_3_ NCs@BaSO_4_ was quenched via the IFE when adding Au NPs. When adding melamine to the above conjugate, the PL emission was restored to the original intensity via the weakening of the IFE. A linear detection range of melamine was recorded between 5 nM and 5 µM with an LOD of 0.42 nM. Through using this method, >95% recoveries of melamine in raw milk samples were achieved with <4% RSDs. This is an inspiring and innovative work for melamine monitoring. Su et al. fabricated a CsPbBr_3_/a-TiO_2_/FTO electrode for the photoelectrochemical immunoassay of aflatoxin B1 (AFB_1_; a carcinogen) by compositing CsPbBr_3_ NCs with amorphous titanium dioxide (TiO_2_) [205]. A linear regression of AFB_1_ detection by CsPbBr_3_/a-TiO_2_/FTO electrode was observed between 32 pM and 48 nM with an LOD of 9 pM. This work was confirmed by recoveries (ranging between 90.2% and 109.0%) in peanut and corn samples; therefore, it can be regarded as one of the most inspiring photoelectrochemical analytical methods.

The employment of CsPbX_3_ (X = C, Br, and I) and composites toward the quantification of pesticides, insecticides, and fungicides has also been demonstrated by many researchers. CsPbBr_3_ QDs (PLQY = 96%), CsPbBr_3_ QD-coated MIPs (PLQY = 92%), CsPbI_3_ QDs (PLQY = 27%), MIP/CsPbBr_3_ QD composites, and MIP-mesoporous silica-embedded CsPbBr_3_ QDs were demonstrated for discriminating ziram (a fungicide), omethoate (an organophosphorus insecticide), clodinafop (an herbicide), phoxim (an organophosphate insecticide), and dichlorvos (2,2-dichlorovinyl dimethyl phosphate; an insecticide), respectively, via PL quenching responses [206,207,208,209,210]. These composites were developed using diverse methods, such as room-temperature-controlled synthesis, the slow hydrolysis of the capping agent, microwave synthesis, self-assembly, etc. It should be noted that cesium lead halide composites have displayed exceptional linear regression with excellent LODs and have been applied in real-time food stuff/soil analysis, as detailed in Table 5. Moreover, all these composites showed negligible interfering effects from competing species, which is rather unique. For example, the MIP/CsPbBr_3_ QD composite [209] displayed greater selectivity than other interferences, as seen in Figure 26. Table 5 summarizes the synthetic route, PLQY, linear range, detection limit, and application of CsPbX_3_ (X = Cl, Br, and I) and composites toward the detection of bioanalytes, drugs, fungicides, and pesticides.

### Critical View on the Detection of CsPbX_3_ (X = Cl, Br, and I)-Based Bioanalytes, Drugs, Fungicides, and Pesticides 

The stability of CsPbX_3_ (X = Cl, Br, and I) in water is the main issue for discriminating bioanalytes, drugs, fungicides, and pesticides [181,182,183,184,185,186,187,188,189,190,191,192,193,194,195,196,197,198,199,200,201,202,203,204,205,206,207,208,209,210]. To avoid the above complications, compositing CsPbX_3_ (X = Cl, Br, and I) with other materials, such as APTES, BN, and MIPs has been proposed. However, there are still a few probes that require critical evaluation in terms of structural degradation in aqueous media. Water-stable CsPbBr_3_/Cs_4_PbBr_6_ NCs were proposed for detecting folic acid [202], but how long can the proposed structure remain stable is still under debate. Another controversial issue regarding CsPbX_3_ (X = Cl, Br, and I) and composite-based biomolecule sensing is how to avoid the toxicity of Pb^2+^ in food sample analysis. Consequently, the sensor’s selectivity to specific analytes requires careful investigation in many reports. Detecting pesticides and fungicides in real samples using CsPbX_3_ (X = Cl, Br, and I) and composites needs critical analysis to overcome the toxicity induced by Pb^2+^.

## 9. Cellular Imaging Applications of CsPbX_3_ (X = Cl, Br, and I) and Composites

Getachew et al. and Kar et al. described the cellular imaging utility of magnesium- and zinc-doped cesium lead halides, namely CsMg_x_Pb_1−x_I_3_ QDs and zinc-doped CsPbBr_3_-Cs_4_PbBr_6_ nanocomposites [211,212]. The doping of metal ions in perovskite quantum dots could result in stability and biocompatibility improvement. In CsMg_x_Pb_1−x_I_3_ QDs, Mg^2+^ partially substituted Pb^2+^ in the CsPbI_3_ framework [211]. When encapsulating CsMgxPb_1−x_I_3_ QDs with gadolinium-conjugated pluronic 127 (PF127-Gd), PQD@Gd nanoagents were achieved, which were applied in ROS detection in cancer cells and photocatalytic studies. The Zn^2+^ doping in CsPbBr_3_-Cs_4_PbBr_6_ NCs (emission at 494–506 nm; PLQY = 88%) improved the stability compared to bare NCs in highly polar solvents [212]. Moreover, the zinc-doped ensemble showed greater biocompatibility, thereby becoming effective in cellular imaging, as visualized in Figure 27. In this report, silica-coated CsPbBr_3_-Cs_4_PbBr_6_@(OA)_2_PbBr_4_ (core–shell NCs synthesized via the LARP method using (3-amino-propyl)trimethoxysilane [APTMS]) without zinc doping were labeled as NC-0. The NCs with 20%, 40%, 60%, and 80% zinc doping were labeled as NC-20, NC-40, NC-60, and NC-80, respectively. NC-40 was engaged in cellular imaging studies because of its exceptional biocompatibility and stability. Similarly, CsPbBr_3_@SiO_2_ core–shell, CsPbBr_3_/SiO_2_/mPEG-DSPE NCs (mPEG-DSPE = polyethylene glycol-grafted phospholipid), and phTEOS-TMOS@CsPbBr_3_ NCs (TMOS and phTEOS represent the alkoxysilanes) were also demonstrated in bioimaging studies [213,214,215]. SiO_2_-coated CsPbBr_3_ core–shell structures reduced the toxicity and made NCs more effective for in vitro cellular imaging [213]. Hydrophobic CsPbBr_3_/SiO_2_ encapsulated with mPEG-DSPE showed better water stability and photostability; thus, they can be applied in multiphoton bioimaging [214]. CsPbBr_3_ NCs coated with alkoxysilanes showed improved stability, water dispersibility, and lower toxicity and thus can be used in two-photon cellular imaging studies [215].

Many research groups reported the preparation of encapsulated CsPbX_3_ (X = C, Br, and I) and composites with a polymer matrix to maintain the structural and emission stability of cesium lead halides. Methoxypolyethylene glycol amine-capped CsPbBr_3_ NCs (CsPbBr_3_/mPEG-NH_2_ NCs), polyvinylidene fluoride (PVDF)-encapsulated CsPbBr_3_, poly-vinyl pyrrolidone (PVP)-capped CsPbX_3_ NCs, poly(lactic-co-glycolic acid) (PLGA)-encapsulated CsPbBr_3_ QDs, and polystyrene-block-poly(acrylic acid) (PS-b-PAA) were developed with high water stability, greater biocompatibility, and low toxicity; thus, they can be applied in long-term cellular imaging studies [216,217,218,219,220]. For cellular imaging of cesium lead halide perovskites, Lou et al. synthesized insoluble CsPbBr_3_/CsPb_2_Br_5_-composited NCs (PLQY = 80%) via water-assisted chemical transformations and HeLa cellular imaging studies [221]. Judging from the synthetic simplicity and low toxicity, this work can be regarded as inspiring. However, more investigations on structural stability and PL emission are required.

### Critical View on CsPbX_3_ (X = Cl, Br, and I)-Based Cellular Imaging

It has been argued that cellular imaging using CsPbX_3_ (X = Cl, Br, and I) could be hindered due to the unstable emission properties of CsPbX_3_ in aqueous media. To avoid the instability issue, capping CsPbX_3_ (X = Cl, Br, and I) with suitable ligands, such as 3-amino-propyl)trimethoxysilane (APTMS), polyethylene glycol-grafted phospholipid (mPEG-DSPE), and alkoxysilanes (TMOS and phTEOS) has been proposed. This capping could not only reduce toxicity due to Pb^2+^ but can also improve biocompatibility [211,212,213,214,215]. However, careful optimization is still required. Polymer capping, metal doping, and surface coating were also used to improve the biocompatibility of CsPbX_3_ (X = Cl, Br, and I) in cellular imaging studies [216,217,218,219,220,221]. Nevertheless, critical assessments of toxicity profiles when applying these composites in cellular imaging still require much attention.

## 10. Advantages and Limitations

The employment of CsPbX_3_ (X = Cl, Br, and I) and composites in sensing investigations has certain advantages and limitations, as noted below.

### 10.1. Advantages

Due to the unique structural features, tuning the photophysical properties of CsPbX_3_ (X = Cl, Br, and I) and anion exchange can result in red, green, and blue emission with an enhanced PLQY (reaches up to 98%); therefore, PL-based sensors with relevant colorimetric responses can benefit from the above properties.The greater carrier mobility of CsPbX_3_ (X = Cl, Br, and I) can be adjusted by combining with other semiconducting materials (such as MoS_2_, graphene, mxenes, etc.), which is advantageous to the fabrication of heterojunction devices and electrodes for photo/chemoresistive and electrochemical detection of a specific analyte [222,223,224,225].The capping and encapsulation of the proposed CsPbX_3_ (X = Cl, Br, and I) system can enhance water stability, which allows for long-term tracking of bioanalytes.Core–shell/polymer encapsulation can produce a low-toxicity, biocompatible CsPbX_3_ (X = Cl, Br, and I) conjugate, which can be applied in bioimaging with comparable performance to other lead-free perovskites and luminescent organic probes [226,227,228,229].Advancing sensing studies in real samples is highly feasible with capped/functionalized CsPbX_3_ (X = Cl, Br, and I) nanostructures; therefore, their performance can be comparable to other nanoprobes (nanoparticles, nanoclusters, nanowires, nanocomposites, etc.) [230,231,232,233,234,235].

### 10.2. Limitations

Many stable CsPbX_3_ (X = Cl, Br, and I) and composites are fabricated using multiple synthetic steps, which require precision optimization. However, optimizing synthesis processes is time-consuming, which restricts the use of CsPbX_3_ (X = Cl, Br, and I) as sensor probes.CsPbX_3_ (X = Cl, Br, and I) and composites can degrade rapidly when exposed to air moisture, elevated temperature, humid conditions, etc., thereby limiting their sensing utility in harsh conditions [236,237,238].The major contributing factor for the sensing performance of CsPbX_3_ (X = Cl, Br, and I) and composites has been attributed to structural/phase transformation [239,240]. Thus, it is questionable if reversible cycles in analyte detection can be realized.To investigate the precise underlying mechanism of the sensing response to a specific analyte, supporting lines of evidence through methods, such as TEM, XPS, dynamic light scattering spectra (DLS), Zeta potential, etc., are necessary. Thus, the cost-effectiveness of research is questionable, which limits such sensor development in developing or underdeveloped countries.Since real water samples may contain certain numbers of ionic species, the reliability of metal ion quantification by CsPbX_3_ (X = Cl, Br, and I) and composites in real samples is still questionable. This limits the application of CsPbX_3_ (X = Cl, Br, and I) probes toward the detection of metal ions and anions.

## 11. Conclusions and Perspectives

This article provides a detailed review of the sensing performance of CsPbX_3_ (X = Cl, Br, and I) and composites toward the detection of analytes, such as metal ions, anions, chemicals, explosives, gases, volatile organic compounds, bioanalytes, humidity, temperature, radiation, etc. In particular, the involved synthetic methods in developing CsPbX_3_ (X = Cl, Br, and I)-based sensor probes, the linear detection range, LODs, and real-time applicability were also tabulated for a broad audience. Discussions on the mechanistic aspects of analyte detection, novelty, deficiencies, and feasible directions were also extrapolated for readers. Finally, the bioimaging applications of CsPbX_3_ (X = Cl, Br, and I)-based composites were illustrated for future biomedical research.

The following questions/perspective points must be addressed or focused on in future research: (A) There are many reports available on the employment of CsPbX_3_ (X = Cl, Br, and I)-based probes toward Cu^2+^ quantification; hence, a state-of-the-art procedure for fabricating commercial sensors must be developed; (B) to date, the detection of Cu^2+^, Hg^2+^, Co^2+^, Zn^2+^, Cd^2+^, UO_2_^2+^, In^3+^, and Fe^3+^ have been demonstrated using CsPbX_3_ (X = Cl, Br, and I)-based probes, but the research direction should be further extended toward the sensing of monovalent (M^+^) cations, divalent (M^2+^) cations, trivalent (M^3+^) cations, etc., by optimizing the functional/capping agents; (C) the method can be further fine-tuned to develop an effective procedure for the commercialization of anion-exchange-facilitated detection of Cl^−^ and I^−^ by CsPbBr_3_ toward real-time monitoring; (D) the optimization of functional/capping moieties can be carried out to extend the sensing capability of CsPbX_3_ (X = Cl, Br, and I)-based probes to existing anions, such as SO_4_^2−^, PO_4_^2−^, P_2_O_7_^2−^, CN^−^, SCN^−^, etc.; (E) the anion-exchange-directed detection of alkyl halides, benzoyl peroxide, and solvent polarity by CsPbX_3_ (X = Cl, Br, and I) and composites requires in-depth investigation to confirm its novelty; (F) advancing CsPbX_3_ (X = Cl, Br, and I) and composites toward the detection of toxic gases, such as carbon monoxide (CO), carbon dioxide (CO_2_), phosgene (COCl_2_), etc., must be the focus in future research; (G) numerous reports on CsPbBr_3_-based NH_3_ sensors are available; therefore, fabricating commercial NH_3_-sensing devices must be the focus in future research; (H) research on humidity, temperature, radiation, and photodetection using CsPbX_3_ (X = Cl, Br, and I) and composites can be expanded to include other nanostructures, such as graphene, metal–organic frameworks, nanosheets, nanotubes, etc.; (I) few reports are available on detecting essential bioanalytes, drugs, fungicides, and pesticides using CsPbX_3_ (X = Cl, Br, and I) and composites; thus, more attention is required in this research direction; (J) in-depth (in vitro/in vivo) cellular imaging, tracking, and the underlying mechanisms of the imaging responses of CsPbX_3_ (X = Cl, Br, and I) and composites and the construction of low-toxicity probes must be carried out in future research; (K) the majority of CsPbX_3_ (X = Cl, Br, and I) and composite-based sensor reports lack theoretical support from density functional theory (DFT) calculations, which should be included in new sensor designs.

Apart from the above open questions and possible future research directions, sensing studies that involve composites of CsPbX_3_ (X = Cl, Br, and I) toward the detection of diverse analytes require more innovative approaches to realize real-time applications. Many scientists are currently working on developing a “state-of-the-art” sensor procedure for the detection of diverse analytes. Thus, innovative breakthroughs can be expected in the near future.

## Figures and Tables

**Figure 1 sensors-24-02504-f001:**
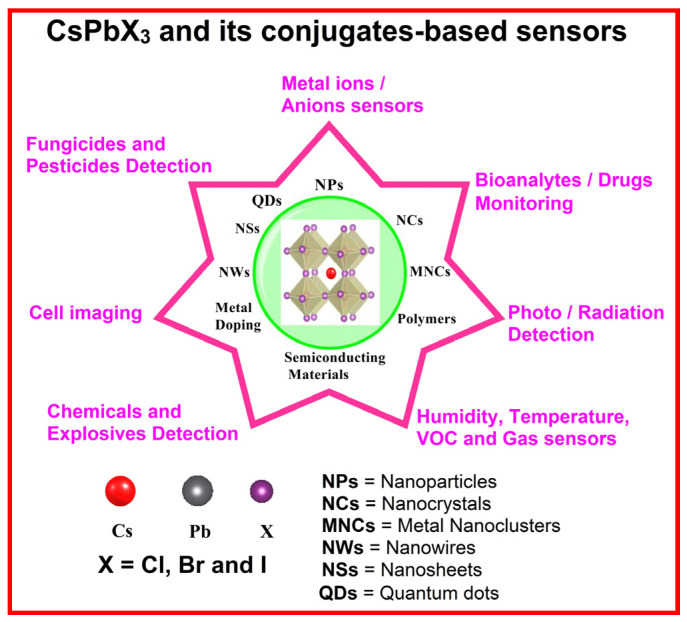
Schematic representation of CsPbX_3_ (X = Cl, Br, and I) and composite-based sensors used in metal ion and anion sensors, bioanalyte and drug monitoring, pesticide detection, cell imaging, etc.

**Figure 2 sensors-24-02504-f002:**
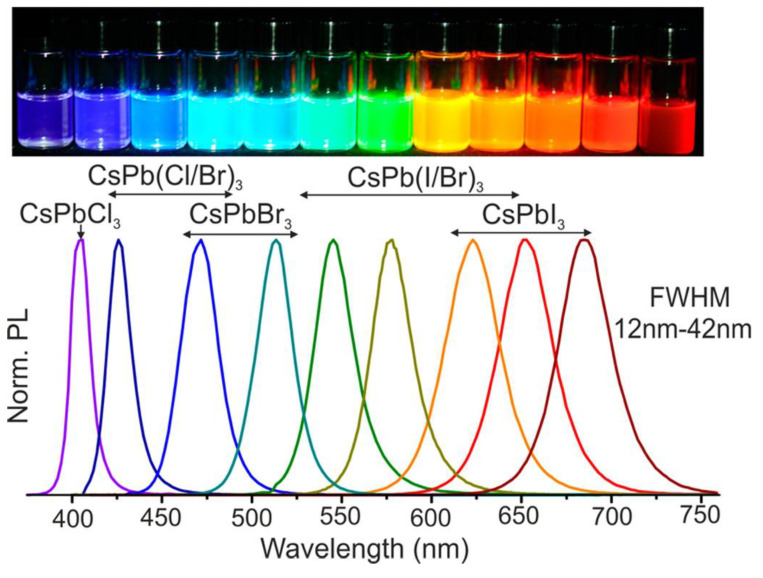
Colloidal perovskite CsPbX_3_ NCs (X = Cl, Br, and I) exhibit size- and composition-tunable bandgap energies covering the entire visible spectral region with narrow and bright emission; colloidal solutions in toluene under UV lamp (λ = 365 nm) is shown in the upper corner of the figure (permission obtained from Ref. [71]).

**Figure 3 sensors-24-02504-f003:**
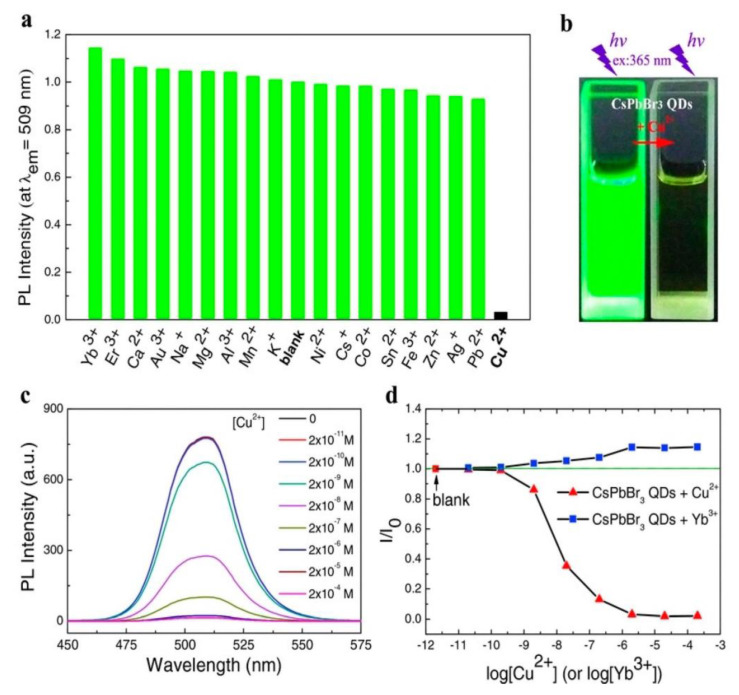
(**a**) The effect of different metal ions on the PL intensity of CsPbBr_3_ QDs. The concentration of metal ions and CsPbBr_3_ QDs are 2.0 × 10^−6^ and ≈1.0 × 10^−9^ M, respectively. The PL peak intensity is normalized by CsPbB_r3_ QDs without adding metal ions (the “blank” column). (**b**) Photo of CsPbBr_3_ QDs in cyclohexane under ultraviolet light excitation with and without Cu^2+^. (**c**) PL spectra of CsPbBr_3_ QDs at different [Cu^2+^] concentrations and (**d**) PL intensity of CsPbBr3 QDs (λ_ex_ = 365 nm) as a function of [Cu^2+^] and [Yb^3+^] (permission obtained from Ref. [81]).

**Figure 4 sensors-24-02504-f004:**
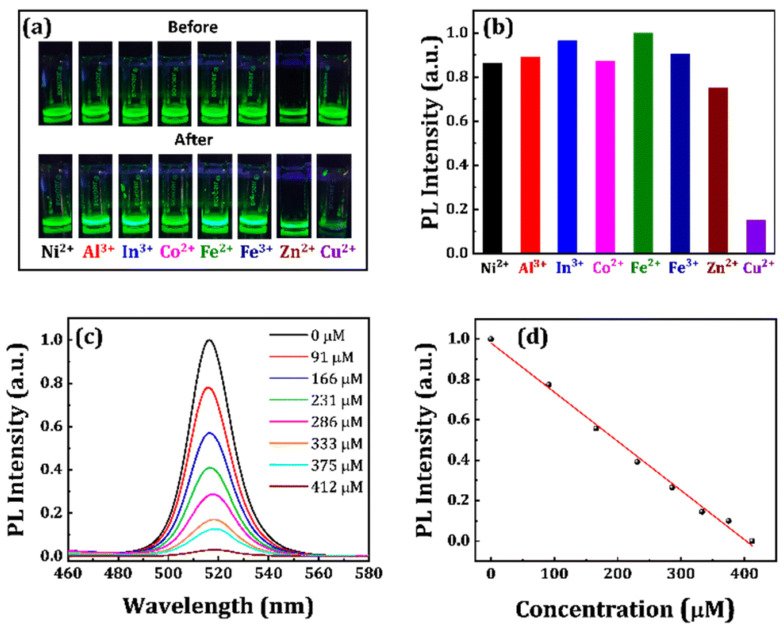
Images of the PbN-4 NC solution under a UV lamp (**a**) before and after adding different metal ions as marked in the figure. (**b**) Chart representing the comparison of the PL intensity of PbN-4 NCs that persisted after the addition of subsequent metal ions. (**c**) Emission spectra of PbN-4 NCs in the presence of different concentrations of Cu^2+^ solutions as shown in the legends. (**d**) The linear curve represents decreasing in the PL intensity of PbN-4 NCs after adding different concentrations of Cu^2+^ solution (permission obtained from Ref. [83]).

**Figure 5 sensors-24-02504-f005:**
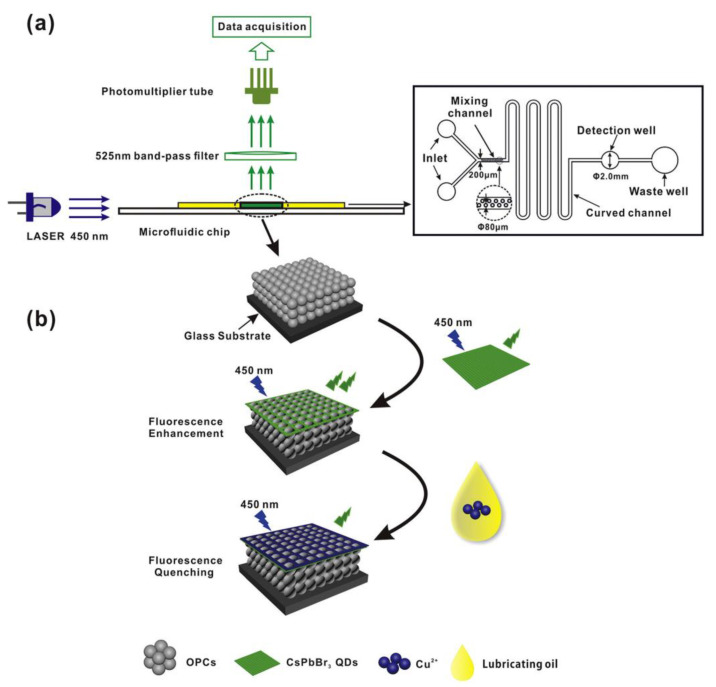
(**a**) The workstation setup for the detection system of microfluidic sensor. (**b**) The schematic of the formation of PMMA OPCs/c composites and Cu^2+^ detection (permission obtained from Ref. [87]).

**Figure 6 sensors-24-02504-f006:**
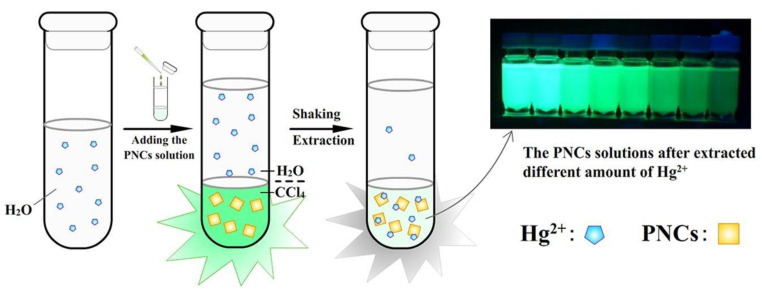
Illustration of liquid–liquid extraction and visual detection of Hg^2+^ using CsPbBr_3_ PNCs (permission obtained from Ref. [92]).

**Figure 7 sensors-24-02504-f007:**
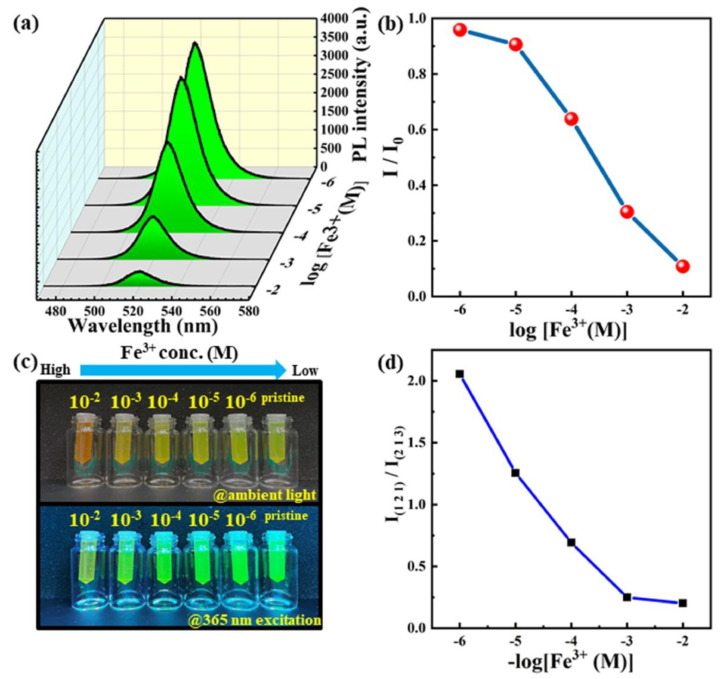
(**a**) Emission spectra and (**b**) normalized emission intensity of AP-PQD in the presence of different Fe^3+^ concentrations. (**c**) Photographs of AP-PQD dispersed in ethanol containing Fe^3+^ under ambient light and 365 nm UV light. (**d**) Intensity ratio of AP-PQD in the presence of different Fe^3+^ concentrations (permission obtained from Ref. [100]).

**Figure 8 sensors-24-02504-f008:**
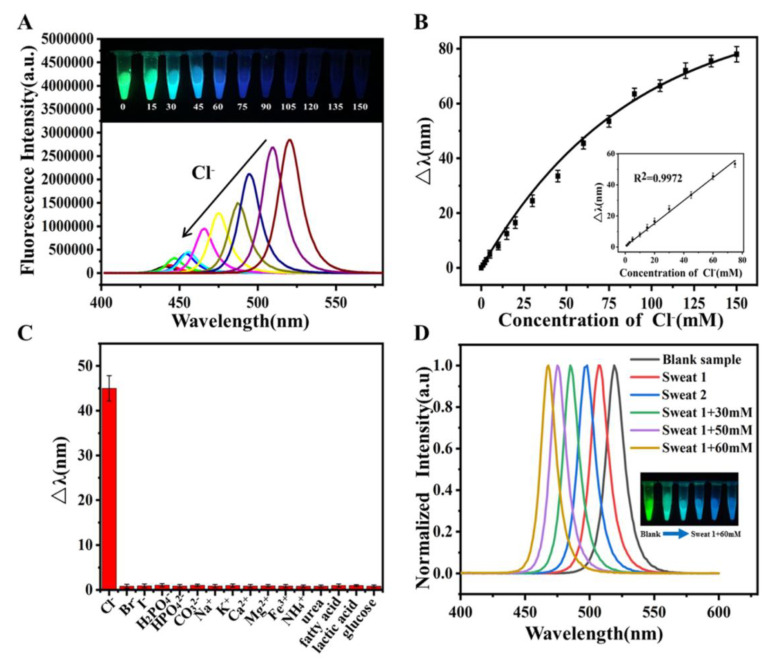
(**A**) Fluorescence spectra and corresponding fluorescence photographs of the CsPbBr_3_/OPA + OAm NCs in the presence of different concentrations of chloride ions from 0 to 150 mM in an aqueous solution under 365 nm UV excitation. (**B**) The fitting curve of Δλ plotted as a function of Cl^−^ concentration; inset: the corresponding calibration curve of Δλ and Cl^−^ concentration from 1 to 80 mM. (**C**) Wavelength shift of different substances for the selectivity investigation of Cl^−^ sensing. (**D**) Fluorescence spectra of actual samples and samples after being spiked with different concentrations of Cl^−^; inset: the corresponding fluorescence photos of the samples (permission obtained from Ref. [103]).

**Figure 9 sensors-24-02504-f009:**
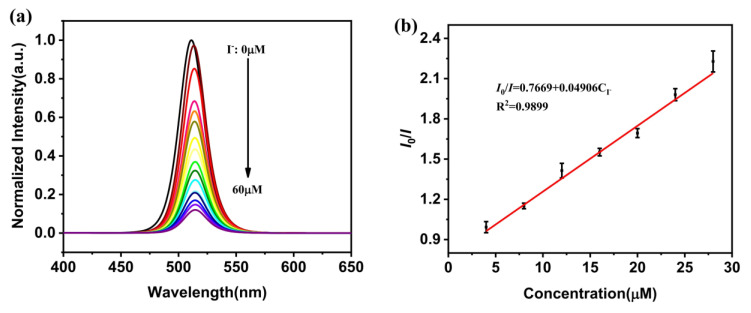
(**a**) Fluorescence spectra of NH_2_-PNCs at various concentrations of I^−^ (0–60 μM). (**b**) Linear fitting curve of I_0_/I of the fluorescence of NH_2_-PNCs and concentrations of I^−^ (permission obtained from Ref. [110]).

**Figure 10 sensors-24-02504-f010:**
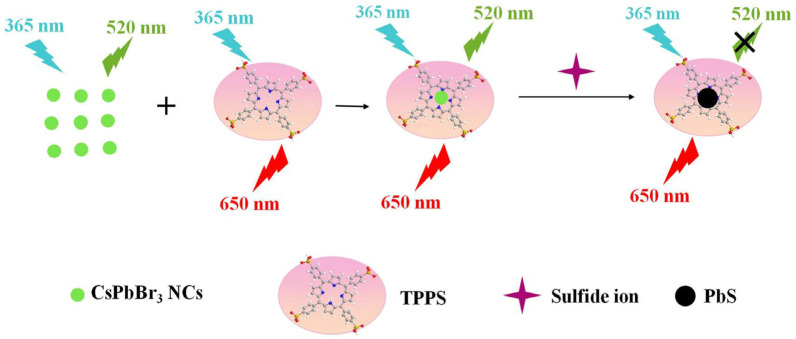
Schematic illustration of CsPbBr_3_/TPPS nanocomposite-based ratiometric fluorescence detection of sulfide ion (permission obtained from Ref. [113]).

**Figure 11 sensors-24-02504-f011:**
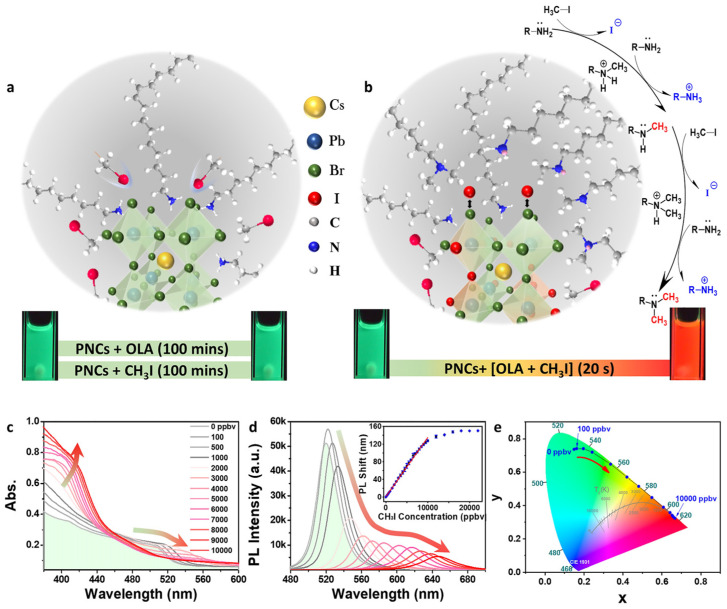
Reaction mechanisms and the spectroscopic response of CsPbBr_3_ perovskite nanocrystals (PNCs) to CH_3_I. (**a**) Oleylamine (OLA, 0.96 mM) or CH_3_I (20,000 ppbv solution) were introduced separately into PNC dispersions in toluene. The emission images under 365 nm UV light were recorded after 100 min, showing no change in emission. (**b**) OLA-pretreated CH_3_I solutions (CH_3_I concentration: 20,000 ppbv) were added to a PNC dispersion in toluene, with the emission color observed under 365 nm UV light before and 20 s after addition. The hypothesized reaction mechanism occurs when CH_3_I induces the alkylation of OLA via the SN^2^ mechanism and stops at dimethyl analog formation. (**c**) UV−visible absorption spectra of PNCs exposed to varying amounts of CH_3_I. (**d**) Emission spectra of CsPbBr_3_ PNCs as a function of the amount of added CH_3_I; inset: redshift of PNC PL emission as a function of CH_3_I concentration. Linear fitting of results from 100 to 10,000 ppbv is shown as a red line with R^2^ = 0.997. (**e**) CIE chart converted from the PL spectra of PNCs exposed to varying amounts of CH_3_I. Note: Spectra in c, d were recorded 20 s after CH_3_I addition at room temperature to ensure the reaction was complete (permission opted from Ref. [114]).

**Figure 12 sensors-24-02504-f012:**
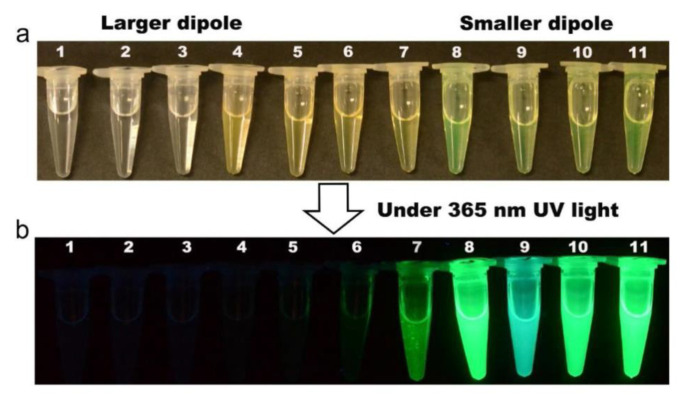
Photos of the effect of different polar solvents on CsPbBr_3_ QD solution. The solvents are arranged basically in the order of increasing dipole moment: (1) DMSO, (2) DMF, (3) methanol, (4) acetonitrile, (5) ethanol, (6) 1-propanol, (7) acetone, (8) ethyl acetate, (9) chloroform, (10) dichloromethane, and (11) toluene; (**a**) under ambient light, (**b**) under 365 nm UV light (permission obtained from Ref. [120]).

**Figure 13 sensors-24-02504-f013:**
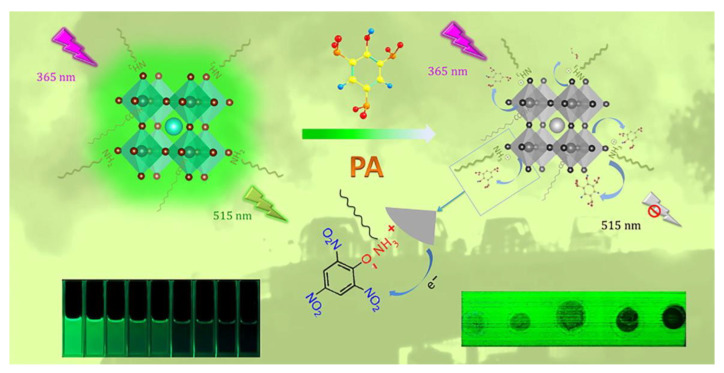
Schematic for the sensitive fluorescence detection of PA based on perovskite quantum dots and its paper-strip applications (permission obtained from Ref. [123]).

**Figure 14 sensors-24-02504-f014:**
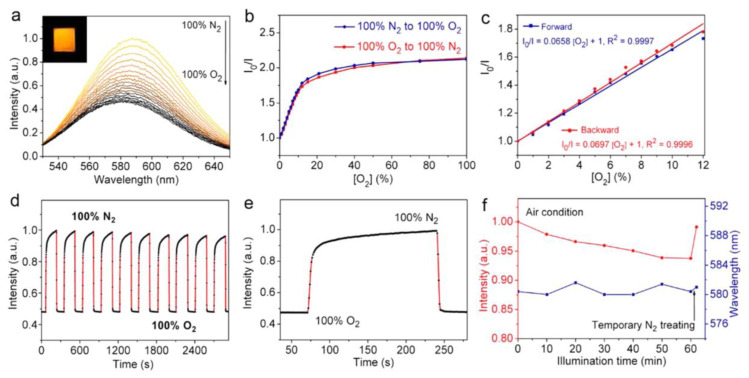
Sensing responses of the Mn_0.175_:CsPb_0.825_Cl_3_ film to O_2_: (**a**) Phosphorescence spectra under different O_2_ fractions (%). (**b**) The first-order reaction kinetics curves of the maximum phosphorescence intensities under different O_2_ fractions (%). (**c**) Stern−Volmer plot under O_2_ fractions between 0 and 12%. (**d**) Reversibility test under the alternating exposure to 100% O_2_ or 100% N_2_ (detected at 586 nm). (**e**) The response time curve within one cycle test. (**f**) Photostability test in air condition (excited at 365 nm) (permission obtained from Ref. [126]).

**Figure 15 sensors-24-02504-f015:**
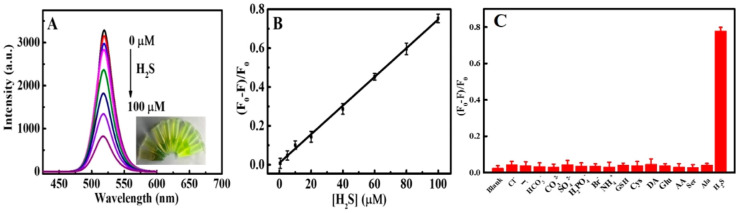
(**A**) Fluorescence spectra of CsPbBr_3_ QDs upon the addition of different concentrations of H_2_S. The concentration of H_2_S from top to bottom was 0−100 μM. (**B**) A linear relationship between the fluorescence intensity of CsPbBr_3_ QDs and H_2_S concentration. (**C**) Changes in fluorescence intensity of CsPbBr_3_ QDs in the presence of H_2_S (100 μM) and other interfering agents (1 mM) (permission obtained from Ref. [133]).

**Figure 16 sensors-24-02504-f016:**
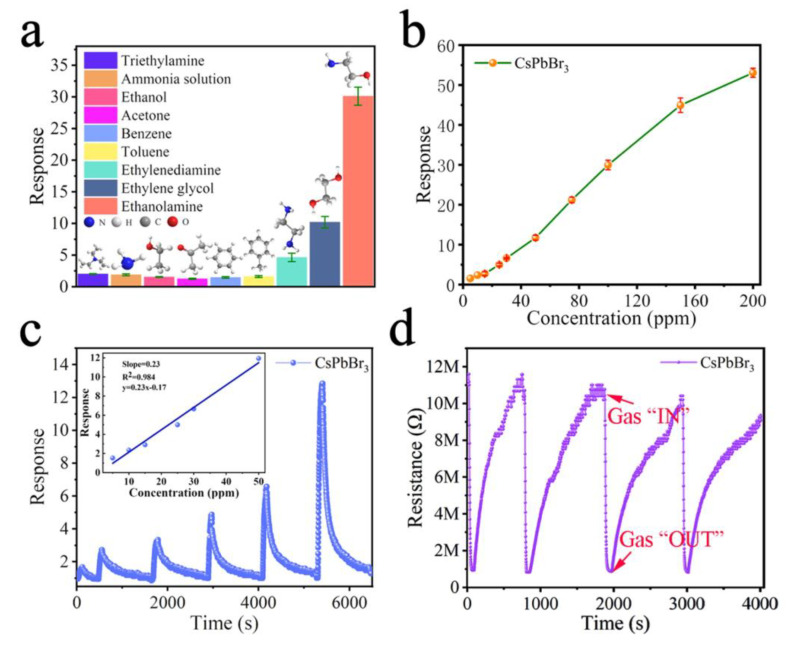
Gas sensitivity test of the CsPbBr_3_ gas sensor at 13% RH: (**a**) Selective response to 100 ppm of different test gases. (**b**) Response curve to different concentrations of EA at RT. The error bars were taken from 5 sets of data. (**c**) Dynamic response curve; the inset shows that the response is linearly related to the low concentration of EA of less than 50 ppm. (**d**) Repeatable response curve of the CsPbBr_3_ sensor to 50 ppm EA at RT (permission obtained from Ref. [141]).

**Figure 17 sensors-24-02504-f017:**
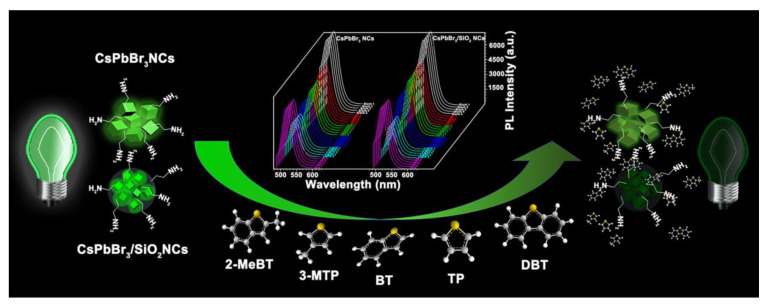
Schematic illustration of the discrimination principle of the fluorescent sensor array for thiophene sulfides based on two perovskite NCs (permission obtained from Ref. [144]).

**Figure 18 sensors-24-02504-f018:**
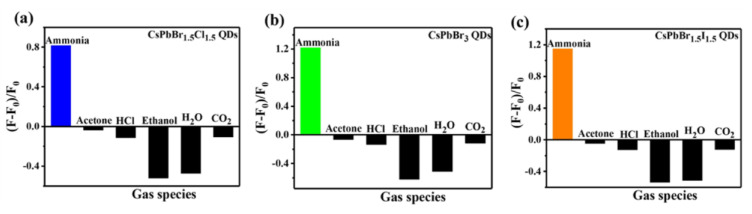
(**a**–**c**) The responses of CsPbBr_1.5_Cl_1.5_, CsPbBr_3_, and CsPbBr_1.5_I_1.5_ perovskite QD film sensors to various gases (permission obtained from Ref. [146]).

**Figure 19 sensors-24-02504-f019:**
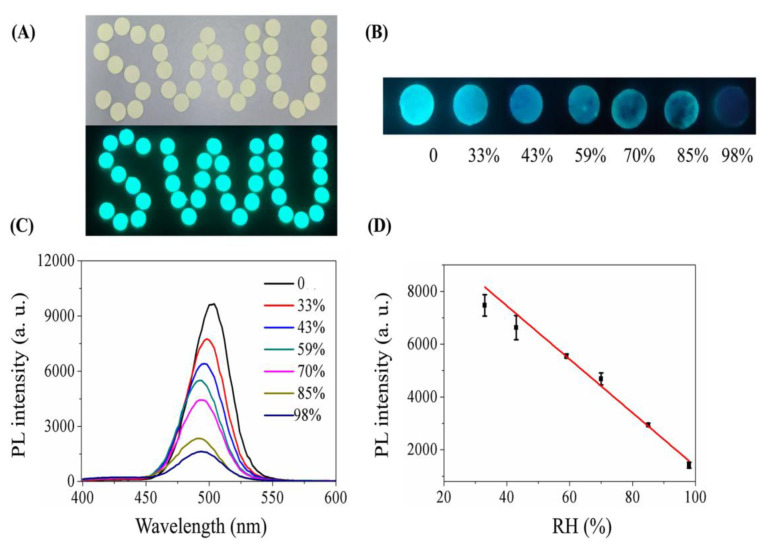
(**A**) PL images of paper substrates loading CsPbBr_3_ perovskite. (**B**) PL images of paper-substrate-loading CsPbBr_3_ exposed to different RHs. (**C**) Humidity-dependent PL spectra of CsPbBr_3_ perovskite loaded on paper substrates. (**D**) Calibration curve for detecting different RHs, n = 3 (permission obtained from Ref. [155]).

**Figure 20 sensors-24-02504-f020:**
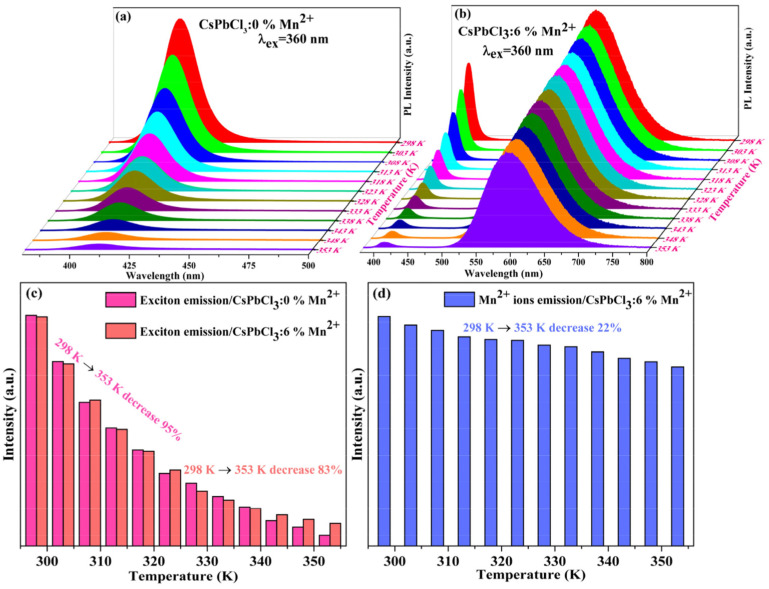
Temperature-dependent emission spectra of (**a**) CsPbCl_3_ and (**b**) CsPbCl_3_:6% Mn^2+^ QDs in the range of 298 K–353 K. The integrated emission intensity centered at (**c**) approximately 410 nm and (**d**) approximately 600 nm as a function of temperature (permission opted from Ref. [160]).

**Figure 21 sensors-24-02504-f021:**
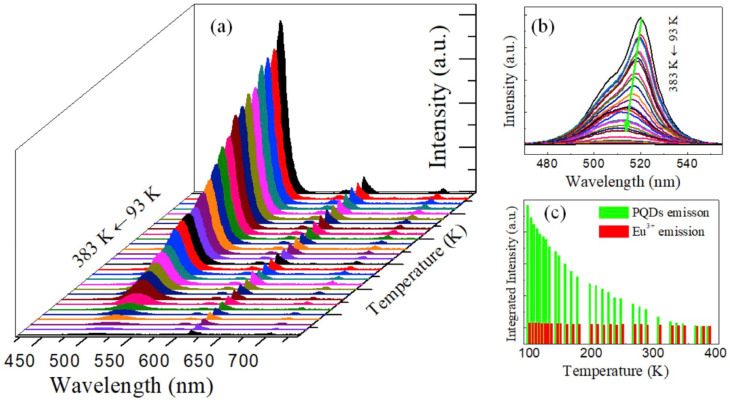
(**a**) Temperature-dependent PL spectra of Eu^3+^-doped PQDs@glass sample (S3). (**b**) Temperature-dependent PL spectra of CsPbBr_3_ PQD emission. (**c**) Integrated emission intensity of CsPbBr_3_ PQDs and Eu^3+^ as a function of temperature (permission obtained from Ref. [164]).

**Figure 22 sensors-24-02504-f022:**
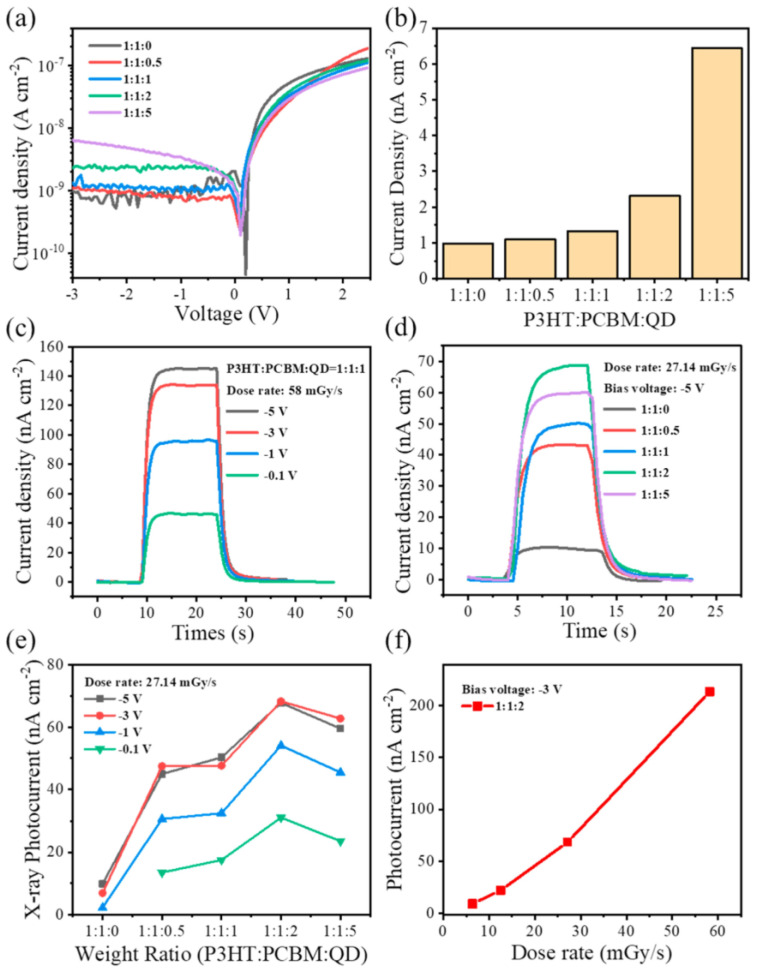
(**a**) Dark J–V curves of the hybrid OPD with different weight ratios of P_3_HT:PC_61_BM:QD. (**b**) Dark current densities of hybrid OPDs under −3 V bias. (**c**) The X-ray response of the device with the weight ratio of 1:1:1 under various bias voltages. (**d**) The X-ray response of the hybrid OPDs with different QD weight ratios (dose rate: 27.14 mGy s^−1^). (**e**) X-ray photocurrent of devices with different QD weight ratios under a reversed bias varying from −0.1 to −5 V. (**f**) The linear response of the OPD device with a weight ratio of 1:1:2 at different dose rates (permission obtained from Ref. [174]).

**Figure 23 sensors-24-02504-f023:**
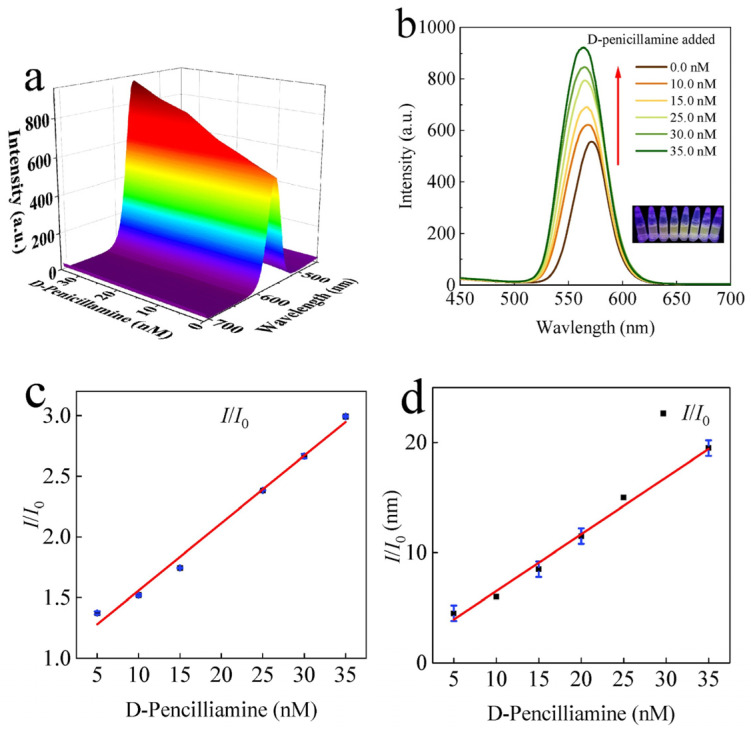
Detection of D-PA by CsPbX_3_ (Br/I) PNCs: (**a**) The 3D photoluminescence spectrum. (**b**) The 2D photoluminescence spectra and photos of corresponding samples at 365 nm (inset). The linear relationship between (**c**) relative intensity of photoluminescence and D-PA concentration, (**d**) photoluminescence wavelength shift, and D-PA concentration (permission obtained from Ref. [187]).

**Figure 24 sensors-24-02504-f024:**
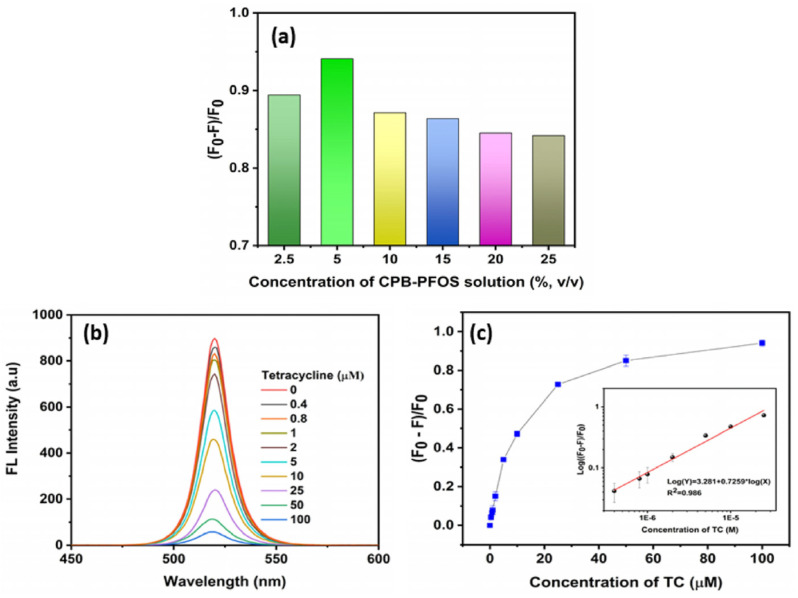
(**a**) Response of FL intensity of different concentrations of CPB-PFOS aqueous solution in the presence of 25 mM of TC, where F and F_0_ are FL intensity with and without TC, respectively. (**b**) The spectra of an aqueous solution of 5% (*v*/*v*) of CPB-PFOS samples with different concentrations of TC under an excitation of 355 nm wavelength. (**c**) Relationship between TC concentration and FL intensity log((F_0_ − F)/F_0_) = 3.281 + 0.7259 × log(C_TC_(M)) with R^2^ = 0.986 (permission opted from Ref. [196]).

**Figure 25 sensors-24-02504-f025:**
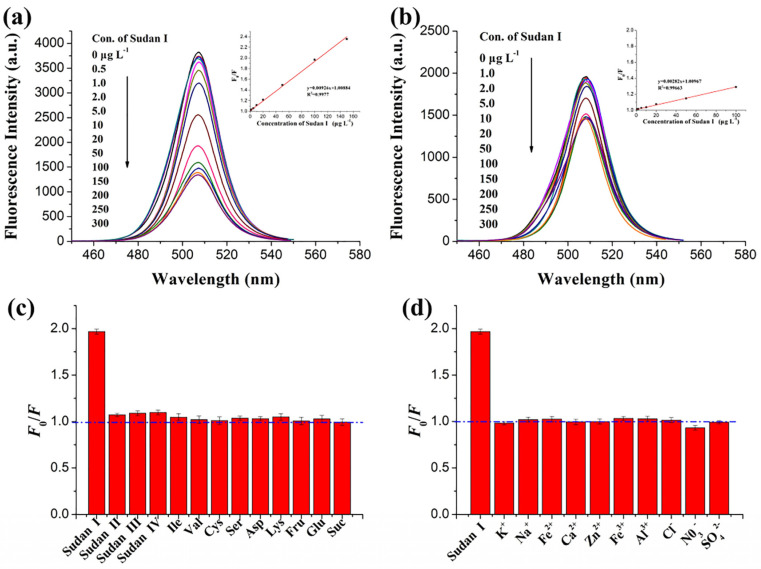
Effect of different concentrations of Sudan I (0–300 µg L^−1^) on the fluorescence spectra of (**a**) MIP-CsPbBr_3_ and (**b**) NIP-CsPbBr_3_ probe; insets of (**a**,**b**) represent the corresponding calibration curve; fluorescence response of MIP-CsPbBr_3_ probe on (**c**) Sudan I, structural analogs, amino acids, and (**d**) some ions (permission obtained from Ref. [203]).

**Figure 26 sensors-24-02504-f026:**
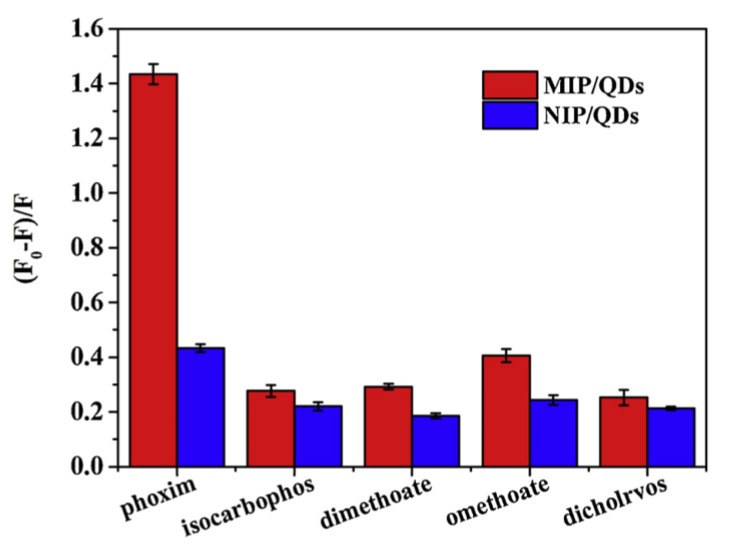
Response performance of the MIP/QDs and NIP/QDs to different OPPs (0.13 μM; OPPS = organophosphorus pesticides) (permission opted from Ref. [209]).

**Figure 27 sensors-24-02504-f027:**
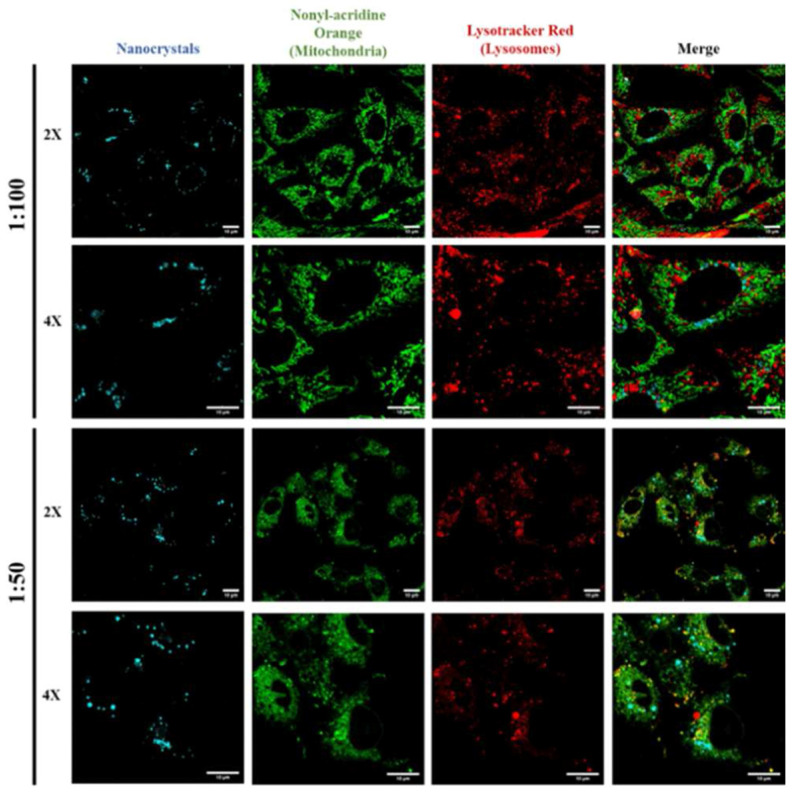
Biocompatibility of NC-40 with mammalian cells. NCs (blue) are internalized by cells and do not associate with mitochondria (green) or lysosomes (red). Higher concentrations of NCs have a detrimental effect on cells as mitochondrial and lysosomal structures become atypical. NCs, nanocrystals (permission obtained from Ref. [212]).

**Table 1 sensors-24-02504-t001:** The synthetic route, PLQY, linear range, detection limit, and application of CsPbX_3_ (X = Cl, Br, and I) and composites toward metal ion detection.

Composition	Synthetic Route; PLQY (%)	Analyte	Method of Detection	Linear Regression	Detection Limit (LOD)	Applications	Ref.
CsPbCl_3_ NCs and CsPbCl_3_ NWs	Hot-injection method; 2.1% and 17.3%	Cu^2+^	PL quenching	0–1 µM	0.06 nM	NA	[80]
CSPbBr_3_ QDs	Hot-injection method; 63%	Cu^2+^	PL quenching	2 nM–2 µM	2 nM	Edible oils	[81]
CSPbBr_3_ QDs	Hot-injection method; 90%	Cu^2+^	PL quenching	0–100 nM	0.1 nM	NA	[82]
Silica-coated CsPbCl_3_ NCs	ligand-assisted reprecipitation (LARP) method; 93%	Cu^2+^	PL quenching	0–412 µM	18.6 µM	Natural water systems	[83]
CsPbBr_3_ QDs	Hot-injection method; NA	Cu^2+^	PL quenching	1 µM–10 mM	NA	NA	[84]
CsPbBr_3_-(SH) polyHIPE composite	Hot-injection method; ~98%	Cu^2+^	PL quenching	10 fM–10 mM	10 fM	NA	[86]
PMMA OPCs/CsPbBr_3_ QD composites	Hot-injection method; NA	Cu^2+^	PL quenching/Microfluidic detection	1 nM–10 mM	0.4 nM	Lubricating oils	[87]
CsPbI_3_ QD/S iO_2_ IOPCs	Hot-injection method; NA	Cu^2+^	PL quenching/Microfluidic detection	0–20 nM and 20–50 nM	0.34 nM	Lubricating oils	[88]
CsPbBr_3_@SiO_2_-E NPs	Hot injection followed by 3-step synthetic modification; 90%	Cu^2+^/S^2−^	PL quenching/PL Recovery	0–5 µM and 5–10 µM (for Cu^2+^) and 0–120 µM (for S^2−^)	0.16 µM (for Cu^2+^) and 8.8 µM (for S^2−^)	NA	[89]
CsPbBr_3_ QD/PMMA fiber membranes	Hot injection/Electrospinning method; 88%	Cu^2+^	PL quenching	1 fM–1 M	1 fM	NA	[90]
CsPbBr_3_@MOF QDs	two step surfactant free procedure; 39.2%	Cu^2+^	PL quenching	100–600 nM	63 nM	NA	[91]
CsPbBr_3_ NCs	Hot-injection method; NA	Hg^2+^	PL quenching	50 nM–10 µM	35.65 nM	NA	[92]
CsPbBr_3_ Crystals	two-step precipitation method; NA	Hg^2+^	PL quenching	5–100 nM	0.1 nM	NA	[93]
CsPbBr_3_-mPEG@SiO_2_ NCs	Ligand engineering and silica encapsulation method; 67.5%	Hg^2+^/GSH	PL quenching/PL Recovery	0.1–50 nM (for Hg^2+^) and 1–10 µM (for GSH)	0.08 nM (for Hg^2+^) and 0.19 µM (for GSH)	Tap water and Serum analysis	[94]
CsPbBr_3_ NCs	Nucleation growth synthesis; >89%	Zn^2+^	PL quenching	0–40 µM	NA	NA	[95]
alpha-amino butyric acid (A-ABA)-capped CsPbBr_3_ QDs (M PQDs)	Hot-injection method; NA	Co^2+^	PL quenching	0–100 nM	0.8 µM	NA	[96]
CsPbBr_3_ QDs	Hot-injection method; NA	UO_2_^2+^	PL quenching	0–3.3 µM	83.33 nM	NA	[97]
PVP shell-grown silica-coated Zn-doped CsPbBr_3_ NCs	Hot-injection method; 88%	In^3+^	PL quenching	0–104 µM	11 µM	NA	[98]
CsPbBr_3_−Ti_3_C_2_Tx MXene QD/QD heterojunction	Hot-injection method; NA	Cd^2+^	PL quenching	99–590 µM	99 µM	NA	[99]
APTES-coated CsPbBr_3_–CsPb_2_Br_5_ QDs	ligand-assisted reprecipitation method; NA	Fe^3+^	PL quenching	10 µM–10 mM	10 µM	NA	[100]

NA = not available; µM = micromole (10^−6^ M); nM = nanomole (10^−9^ M); fM = femtomole (10^−15^ M).

**Table 2 sensors-24-02504-t002:** The synthetic route, PLQY, linear range, detection limit, and application of CsPbX_3_ (X = Cl, Br, and I) and composites toward the detection of anionic species.

Composition	Synthetic Route; PLQY (%)	Analyte	Method of Detection	Linear Regression	Detection Limit (LOD)	Applications	Ref.
CsPbBr_3_ nanoplatelets	Hot-injection method; 83.7%	Cl^−^ and As^3+^	PL peak shift	0.2–0.4 nM and 6.4–58 nM	28 pM and 1 nM	NA	[101]
CsPbBr_3_ QDs	Hot-injection method; 87%	Cl^−^	PL peak shift	10–200 µM	4 µM	Real-time water analysis	[102]
CsPbBr_3_ NCs	Ligand-assisted synthesis; >40%	Cl^−^	PL peak shift	1–80 mM	0.34 mM	Human sweat sample analysis	[103]
CsPbBr_3_ NCs	Hot-injection method; NA	Cl^−^	PL peak shift	10–130 mM	3 mM	Human sweat sample analysis	[104]
CsPbBr_3_ NCs	Hot-injection method; NA	Cl^−^	PL peak shift	100 µM–10 mM	100 µM	Glass/Paper-strip analysis	[105]
β-cyclodextrin stabilized, Arginine added CsPbBr_3_ NCs (ACD-PNCs)	Ligand-assisted synthesis; 82%	Cl^−^ and I^−^	PL peak shift	0.04–0.8 mM and 0.04–1.16 mM	3.2 µM and 9 µM	Human saliva, sweat, and test-strip analysis	[106]
CsPbBr_3_@SiO_2_ NCs	Room-temperature synthesis; NA	Cl^−^	PL peak shift	0–3%	0.05 mg/g	Sand Analysis	[109]
NH_2_-functionalized CsPbBr_3_ NCs	Hot-injection method; NA	Cl^−^	PL Quenching	4–28 µM	1 µM	NA	[110]
CsPbBr_3_ QD/Cellulose composite	Hot-injection method; NA	Cl^−^ and I^−^	PL peak shift	0.1 mM–1 M	2.56 mM (For Cl^−^) and 4.11 mM (For I^−^)	Real-time water analysis	[111]
Tetraphenylporphyrin tetrasulfonic acid (TPPS)-modified CsPbBr_3_ NCs	Hot-injection method followed by compositing; NA	S^2−^	PL Quenching	0.2–15 nM	0.05 nM	Real-time water analysis	[113]

NA = not available; µM = micromole (10^−6^ M); nM = nanomole (10^−9^ M); mM = millimole (10^−3^ M); mg = milligram; g = gram.

**Table 3 sensors-24-02504-t003:** The synthetic route, PLQY, linear range, LOD, and application of CsPbX_3_ (X = Cl, Br, and I) and composites toward the detection of chemicals and explosives.

Composition	Synthetic Route; PLQY (%)	Analyte	Method of Detection	Linear Regression	Detection Limit (LOD)	Applications	Ref.
CsPbBr_3_ NCs	Hot-injection method; NA	CH_3_I	PL peak shift	0.7–70 µM	0.2 ± 0.07 µM	NA	[114]
Yttrium single-atom-doped CsPbBr_3_ NCs	Hot-injection method; NA	CH_3_I	PL peak shift	5.6–157 µM	0.3 µM	NA	[115]
CsPbX_3_ (X = Cl, Br, or I) NCs	Hot-injection method; NA	CH_2_Cl_2_ and CH_2_Br_2_	PL peak shift	0–0.9 M and 7.2–21 mM	48 mM and 1.7 mM	Microfluidic application	[116]
CsPbBr_3_ NCs	NA	Benzoyl peroxide	Peak shift and Ratiometric detection	NA	NA	Food sample analysis	[118]
CsPbBr_3_ NCs	Hot-injection method; 87%	Benzoyl peroxide	Peak shift and Ratiometric detection	0 µM–120 µM	0.13 µM	Food sample analysis	[119]
CsPbBr_2_I microcrystals	Hot-injection method; NA	Nitrophenol	PL Quenching	0.1–0.6 mM	NA	NA	[122]
CsPbBr_3_ and CsPbI_3_ QDs	Hot-injection method; 52.88% and 46.18%, respectively	Picric acid	PL Quenching	0–180 nM and 0–270 nM, respectively	0.8 nM and 1.9 nM, respectively	Paper-strip analysis	[123]

NA = not available; µM = micromole (10^−6^ M); nM = nanomole (10^−9^ M); mM = millimole (10^−3^ M).

**Table 4 sensors-24-02504-t004:** The synthetic route, PLQY, linear range, LOD, and application of CsPbX_3_ (X = Cl, Br, and I) and composites toward the detection of gas and VOCs.

Composition	Synthetic Route; PLQY (%)	Analyte	Method of Detection	Linear Regression	Detection Limit (LOD)	Applications	Ref.
Mn:CsPbCl_3_ NCs	Heat-up strategy; NA	O_2_	PL quenching	0–12%	NA	NA	[126]
CsPbBr_3_ NFs	Hot-injection method; NA	N_2_	PL quenching	1–20 ppm	1 ppm	NA	[128]
CsPbBr_3_ QDs	Sonication Method; NA	H_2_S	PL quenching	0–100 µM	0.18 µM	Rat brain studies	[133]
CsPbBr_3_@CMO	Sonication followed by compositing method; NA	H_2_S	PL quenching	0.15–105 µM	53 nM	Rat brain studies	[134]
CsPbBr3@SBE-β-CD nanocomposite	Sonication followed by compositing method; NA	H_2_S	PL quenching	0.5 µM–6 mM	0.19 µM	Zebrafish studies	[136]
CsPbBr3/NCM composite	Hot injection followed by EDC−NHS method; NA	tripropylamine (TPrA) and Cesium oleate	Electrochemiluminescence (ECL) signals	10 mM and NA	NA and 1 aM	NA	[143]
CsPbBr_3_/SiO_2_ NCs	Precursor injection followed by compositing method; NA	Thiophene Sulfides	PL quenching	10–50 ppm	≈10 ppm	NA	[144]
CsPbBr_3_ QD film	Hot-injection method; NA	NH_3_	PL enhancement	25–300 ppm	8.85 ppm	Film-based sensor study	[145]
CsPbX_3_ (X = Cl, Br, I or mixed halogen) QD film	Hot-injection method; NA	NH_3_	PL enhancement	25–200 ppm	≈20 ppm	Film-based sensor study	[146]
CsPbBr_3_–SiO_2_ nanocomposites on PVDF membrane	controllable strategy; NA	NH_3_	PL quenching	2160–3600 ppm	NA	Test paper study	[147]
CsPbBr_3_ NFs	Hot-injection method followed by electrospinning; NA	NH_3_	PL quenching	528 µM–1.76 mM	<0.5 mM	NA	[148]
CsPbBr_3_/BNNF composites	Hot-injection method followed by compositing method; ~54%	NH_3_	PL quenching	NA	NA	NA	[149]
CsPbX_3_ (X = Cl, Br, and I) QDs @Fe/X-n	Hydrothermal crystallization followed by in situ growth of QDs; NA	NH_3_	PL quenching	0–10 mL	NA	NA	[150]

NA = not available; mM = millimole (10^−3^ M); µM = micromole (10^−6^ M); nM = nanomole (10^−9^ M); aM = attomole (10^−18^ M); ppm= parts per million.

**Table 5 sensors-24-02504-t005:** The synthetic route, PLQY, linear range, detection limit, and application of CsPbX_3_ (X = Cl, Br, and I) and composites toward the detection of bioanalytes, drugs, fungicides, and pesticides.

Composition	Synthetic Route; PLQY (%)	Analyte	Method of Detection	Linear Regression	Detection Limit (LOD)	Applications	Ref.
CsPbBr_3_@Cu nanohybrid	In situ synthesis; NA	H_2_O_2_ and Glucose	Ratiometric PL response	0.2–100 µM and 2–120 µM	0.07 µM and 0.8 µM	Human serum analysis	[181]
TiO_2_/CsPbBr_1.5_I_1.5_ composite film	Slow volatilization method; NA	Dopamine (DA)	Photoelectrochemical (PEC) detection	0.1–250 μM	12 nM	Human serum analysis	[182]
TiO_2_ IOPCs/CsPbCl_3_	slow volatilization method; NA	alpha-fetoprotein (AFP)	Photoelectrochemical (PEC) detection	0.08–980 ng/mL	30 pg/mL	NA	[183]
CsPbBr_3_ microcrystals	one-pot synthesis method; 60%	Uric acid (UA)	PL quenching	3.1 nM–1.33 µM	0.063 ppm	Human blood serum analysis	[184]
CsPbBr_3_ NC-TPPS nanocomposite	Self-assembly strategy; 60%	Acetylcholinesterase (AChE)	PL quenching	0.05–1.0 U/L	0.0042 U/L	Human serum analysis	[185]
CsPbX_3_ (X = Br/I) PNCs	Anion exchange method; NA	Penicillamine	PL enhancement	5.0–35.0 nM	1.19 nM and 5. 47 nM	NA	[187]
CsPbBr_3_ QD-DNA/MoS_2_ NS	One-pot synthetic method; NA	Mycobacterium tuberculosis (Mtb)	PL enhancement	0.2–4.0 nM	51.9 pM	Clinical tuberculosis pathogen analysis	[188]
CsPbBr_3_ NCs@PL	Film hydration method; NA	Pore-forming biotoxins	PL quenching	50 nM–150 µM	50 nM	Bacterial study	[191]
Phospholipid-coated CsPbBr_3_ NCs	Film hydration method; NA	Prostate-specific antigen (PSA)	PL enhancement and colorimetric sensing	0.01–80 ng/mL and 0.1–15 ng/mL	0.081 ng/mL and 0.29 ng/mL	Clinical sample analysis	[192]
Apt-PNCs@cDNA-MNPs	One-pot synthesis, magnetic stirring, and sonication; NA	Peanut allergen Ara h1	PL enhancement	0.1–100 ng/mL	0.04 ng/mL	Food sample analysis	[193]
APTES-functionalized CsPbBr_3_ QDs	Slow hydrolysis of the capping agent; 46.86%	Tetracycline	PL quenching	0.5–15.0 µM	76 nM	Soil sample analysis	[194]
LMSNs@CsPbBr_3_ QDs	Water emulsion followed by homogeneous mixing; ~54%	Tetracycline	PL quenching	0.7–15 µM	93 nM	Water sample analysis	[195]
Cs_4_PbBr_6_/CsPbBr_3_ NPs	Temperature-controlled synthesis; NA	Tetracycline	PL quenching	0.4–10 µM	76 nM	Food sample analysis	[196]
Molecularly imprinted CsPbBr_3_ QDs	Water emulsion followed by homogeneous mixing; NA	Tetracycline	PL quenching	0.2–5 µM	28 nM	Water sample analysis	[197]
CsPbBr_3_@BN	Hot injection followed by calcination; NA	Tetracycline	PL quenching	0–99 µM	14.6 µM	Honey and milk samples analysis	[198]
CsPbBr_3_ NCs	Hot-injection method; NA	Ciprofloxacin hydrochloride	PL peak shift	0.8–50 mM	0.1 mM	Paper-strip analysis	[199]
CsPbB_r3_-loaded MIP nanogels	In situ hot-injection method; NA	Roxithromycin	PL quenching	100 pM–100 nM	20.6 pM	Animal-derived food product analysis	[200]
CsPbBr_3_ QDs	ligand-assisted reprecipitation method; 42%	Cefazolin	Chemiluminescence	25–300 nM	9.6 nM	Human plasma, urine, water, and milk samples analysis	[201]
Water-stable fluorescent CsPbBr_3_/Cs_4_PbBr_6_ NCs	Water emulsion method; NA	Folic acid	PL quenching	10–800 µM	1.695 µM	Urine sample analysis	[202]
MIP-CsPbX_3_ (X = Cl, Br, and I) fluorescent-encoding microspheres	Encoding of MIPs with CsPbX_3_ QDs: NA	Sudan I	PL quenching	2–604 nM	1.21 nM	Food sample analysis	[203]
CsPbBr_3_ NCs@BaSO4	Aqueous emulsion process; 80.3%	Melamine	PL enhancement	5 nM–5 µM	0.42 nM	Spiked dairy sample analysis	[204]
CsPbBr_3_/a-TiO_2_/FTO	Hot injection followed by compositing; NA	Aflatoxin B1 (AFB_1_)	Photoelectrochemical immunoassay	32 pM–48 nM	9 pM	Food sample analysis	[205]
CsPbBr_3_ QDs	Room-temperature-controlled synthesis; 96%	Ziram	PL quenching	0.1–50 ppm	0.086 ppm	Food sample analysis	[206]
CsPbBr_3_ QDs coated MIPs	Slow hydrolysis of the capping agent; 92%	Omethoate	PL quenching	0–1.9 µM	88 nM	soil and cabbage samples analysis	[207]
CsPbI_3_ QDs	Microwave synthesis; 27%	Clodinafop	PL quenching	0.1–5 μM	34.7 nM	Food sample analysis	[208]
MIP/CsPbBr_3_ QD composite	Hot injection followed by self-assembly method; NA	Phoxim	PL quenching	16.8–335.4 nM	4.9 nM	Food sample analysis	[209]
MIP-mesoporous silica-embedded CsPbBr_3_ QDs	Multiple synthetic methods; NA	Dichlorvos	PL quenching	23–110 nM	5.7 nM	Food sample analysis	[210]

NA = not available; mM = millimole (10^−3^ M); µM = micromole (10^−6^ M); nM = manomole (10^−9^ M); pM = picomole (10^−12^ M); ng = microgram (10^−9^ g); pg = picogram (10^−12^ g); ppm = parts per million.

## Data Availability

Not applicable.

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
