# Peer review of "Sensing Utilities of Cesium Lead Halide Perovskites and Composites: A Comprehensive Review"

_sensors, 2024, doi:10.3390/s24082504_

Round 1

Reviewer 1 Report

Comments and Suggestions for Authors

Review paper title “Sensing Utilities of Cesium Lead Halide Perovskites and Conjugates: A Comprehensive Review” provides of sensing utilities of CsPbX3 and conjugates, in the quantitation of metal ions, anions, chemicals, explosives, bio-analytes, pesticides, fungicides, cellular imaging, volatile organic compounds (VOCs), toxic gases, humidity, temperature, radiation, and photodetection. Many literatures were reviewed and commented. This paper was favorable for promoting the development of efficient sensor. It is recommended to be published in Sensors after minor revision.

-Some references need to be corrected.

-Author should discuss and cite Figure 3 in text.

-QDs should be inserted in Figure 1.

-Author should discuss about superior properties of QDs, NCs, MNCs, NWs, NSs, and NPs in introduction in different paragraph.

-Author should give their own opinion on “Relation between PLQY and QDs, NCs, MNCs, NWs, NSs. For example: CsPbCl3 NCs shown PLQY 2.1% where as CsPbCl3 NWs shown PLQYs 17.3%, Why?

Comments on the Quality of English Language

The language and grammar in this paper need polish

Author Response

Reviewer 1

Review paper title “Sensing Utilities of Cesium Lead Halide Perovskites and Conjugates: A Comprehensive Review” provides of sensing utilities of CsPbX3 and conjugates, in the quantitation of metal ions, anions, chemicals, explosives, bio-analytes, pesticides, fungicides, cellular imaging, volatile organic compounds (VOCs), toxic gases, humidity, temperature, radiation, and photodetection. Many literatures were reviewed and commented. This paper was favorable for promoting the development of efficient sensor. It is recommended to be published in Sensors after minor revision.

We are very grateful to the reviewer for providing the opportunity to improve the standard of this review article.

-Some references need to be corrected.

“Author Response”

As suggested by the reviewer, we have corrected many references.

-Author should discuss and cite Figure 3 in text.

“Author Response”

Figure 3 is already cited in the text, please see the highlighted text in lines 248 – 253.

-QDs should be inserted in Figure 1.

“Author Response”

As recommended by the reviewer, the QDs are inserted in Figure 1.

-Author should discuss about superior properties of QDs, NCs, MNCs, NWs, NSs, and NPs in introduction in different paragraph.

Author Response”

As recommended by the reviewer, the superior properties of QDs, NCs, MNCs, NWs, NSs, and NPs and their importance in sensor designs are explored in the introduction as follows.

“The electro-optical properties and sensing responses of CsPbX3 and composites may vary depending on their distinct nanostructures. For example, the CsPbX3-based sensing probes/composites in QD structures with various sizes may possess diverse bandgaps and display red-to-blue wide optical properties [34], which allows the design of dual-mode sensors. Subsequently, the CsPbX3 nanocrystals (NCs) also display unique magnetic and optoelectronic properties. The facile synthesis of NCs allows them to be adopted in distinct applications, such as solar cells and in-vitro/in-vivoapplications [35]. Due to their structural features, such as hardness, diffusivity, density, enhanced ductility/toughness, elasticity, and conductivity/thermal properties, the CsPbX3 NCs can be effectively applied in energy-related studies. For example, Hu et al. defined the use of CsPbBr3 NCs as single photon emitters [36]. Metal nanoclusters (MNCs) showed exceptional physicochemical properties, such as surface modifiability, surface-to-volume ratio, number of atoms, biocompatibility, photothermal stability, etc [37], thereby conjugating with the CsPbX3 may enhance performance of the target-specific sensors. Because the reduced dimensionality of nanowires (NWs) can significantly improve the electric/heat transport than that of the bulk, thus they have great potential in making temperature and chemoresistive sensors [38,39]. For instance, Zhai and co-workers reported the solvothermal synthesis of CsPbX3 (X = Cl, Br) NWs and demonstrated them in photodetector applications [40]. Regarding the two-dimensional materials, the nanosheets (NSs) were demonstrated as effective sensors due to their exceptional physical, chemical, optical, mechanical, electronic, and magnetic properties [41]. Lv et al. demonstrated the generalized colloidal synthesis of two-dimensional cesium lead halide perovskite nanosheets for applying in photodetector applications [42]. Furthermore, nanoparticles with high surface to volume ratios were employed in multiple sensors, which can be operated at distinct solvent environment and elevated temperatures [43]. Based on above reasons, the CsPbX3 probes/composites derived from QDs, NCs, MNCs, NWs, NSs, and NPs require detailed review.”

.”  

-Author should give their own opinion on “Relation between PLQY and QDs, NCs, MNCs, NWs, NSs. For example: CsPbCl3 NCs shown PLQY 2.1% whereas CsPbCl3 NWs shown PLQYs 17.3%, Why?

Author Response”

As endorsed by the reviewer, it has been justified as follows.

“The PLQY depended on the size of the nanostructures (due to band gap variations), which can be adjusted with metal doping. The doping of Yb3+ in 1D-CsPbCl3 NWs showed the higher aspect ratio, uniformity and lower number of defects than that of the undoped ones, thus the Yb-doped 1D-CsPbCl3 NWs showed the high PLQY. The enhanced defective and rough surface morphologies of 1D-CsPbCl3 NCs effectively hindered the light absorption and electron/charge transport, thereby lowing the PLQY.”

Reviewer 2 Report

Comments and Suggestions for Authors

·         The word sensory is not used always correctly in the manuscript, in many cases it should be replaced with the words sensors or sensing.

·         In the keywords the word biosensors should be removed. A biosensor is material with a biomolecule immobilized on its surface that is used to determine the concentration of an analyte. A biosensor is not simply a material that senses a biomolelule or an organic molecule etc as it is used in this report.

·         In the introduction I cannot see clearly which are other perovskites that have been used for the same sensing applications and why Cesium Lead Halide perovskites are better.

·         One of the main problems in the use of perovskites as sensors is their poor stability in water or aqueous solutions. This is mentioned but particularly in sections 8 and 9 this is not mentioned in all cases described and is very important when developing bioanalytical sensors.

·         In section 8 there is also no clear mention about the stability and the selectivity of these sensors. The same perovskite is used for the determination of H2O2 and glucose or for dopamine and uiric acid. Most of the times in biological samples we don’t want the material to sense all of them (poor selectivity). This should be discussed. Or the detection mechanisms are hardly described and this should be commented as well.

·         Lots of typos, or unnecessary gaps, lines 1442, 544, 550, 621,732, (lines 954 and 959 the sign of degrees Celsius is not the same).

·         In line 49 of the introduction there is mention of among all inorganic perovskites, please be more specific which are these perovskites and why the one selected in this review paper is better. In lines 68 and 69, the authors mention combining perovskites with suitable/proper nanomaterials, please add references and mention a few of them.

·         Almost all of the sensors mentioned are based on PL measurements, please mention in the introduction if Cesium Lead Halide perovskites have been used for the development of other type of sensors, electrochemical etc. and add the necessary references.

·         In section 3 please do not start by showing a figure. It should be placed later on.

·         Generally speaking when you describe the sensors developed by many researchers, do not just mention what they did, be more critical and comment on these. Try to comment on the mechanism or why a mechanism of sensing is not given. What are the problems with selectivity stability etc. and studies in real samples and how the researches did overcome them. By going from one sensor to the other, especially if they are developed for the same analyte, try to compare them.

·         In table one most of the sensors listed there are for copper ions.  Also, the method used for the synthesis of the perovskite is most of the times hot injection. Try first to explain why such an interest for these particular metal ions and especially copper and also which methods have been used mostly for the synthesis of the perovskites and what are the advantages of each method.

Comments on the Quality of English Language

 Lots of typos, or unnecessary gaps, lines 1442, 544, 550, 621,732, (lines 954 and 959 the sign of degrees Celsius is not the same). Minor editing of English language required.

Author Response

Reviewer 2

We thank the reviewer for the valuable comments, which allow us to enhance our review article. We have followed the reviewer’s suggestion and rectified the pointed issues.

  The word sensory is not used always correctly in the manuscript, in many cases it should be replaced with the words sensors or sensing.

Author Response”

As per reviewer suggestion, the word sensory is replaced by sensors or sensing in the entire manuscript.

In the keywords the word biosensors should be removed. A biosensor is material with a biomolecule immobilized on its surface that is used to determine the concentration of an analyte. A biosensor is not simply a material that senses a biomolelule or an organic molecule etc as it is used in this report.

Author Response”

As suggested by the reviewer, the key word “Biosensors” is replaced by “Bioanalytes quantitation”

In the introduction I cannot see clearly which are other perovskites that have been used for the same sensing applications and why Cesium Lead Halide perovskites are better.

Author Response”

This has been rectified in the revised version in lines 48 – 49.

One of the main problems in the use of perovskites as sensors is their poor stability in water or aqueous solutions. This is mentioned but particularly in sections 8 and 9 this is not mentioned in all cases described and is very important when developing bioanalytical sensors.

Author Response”

We agree that perovskites have stability issues. So, we deliver the role of structural stability in sensors in section “2. Role of structural stability and optoelectronic properties in sensors”. Further, additional lines are also included in this section with proper citations as noted below.

“The stability of hybrid halide perovskites also follows a similar trend as the CsPbX3-based materials [9]. However, the metal oxide perovskites show slightly better stability than that of the CsPbX3 and hybrid halide perovskites [9–12].”

Moreover, we feel that mentioning the stability of perovskites-based sensors everywhere in sections 8 and 9 is unnecessary. Repeatedly mentioning those statements will affect the readability/citation metrics of our review, and new sensor designs. We appreciate the reviewer’s understanding.

  In section 8 there is also no clear mention about the stability and the selectivity of these sensors. The same perovskite is used for the determination of H2O2 and glucose or for dopamine and uiric acid. Most of the times in biological samples we don’t want the material to sense all of them (poor selectivity). This should be discussed. Or the detection mechanisms are hardly described and this should be commented as well.

Author Response”

We thank the reviewer for the expertise question. We feel that mentioning the stability of perovskites-based sensors everywhere in section 8 is not necessary, which will affect the readability/citation of our review, and new sensor designs. Explanations of detection mechanisms were illustrated wherever required. The scope of this review is to state the importance of CsPbX3-based sensors, and the progress with certain mechanisms. For comparison, we have also delivered tables.  Please, do not ask us to over-emphasize the stability issue only. Hope the reviewer understands the situation.  

Lots of typos, or unnecessary gaps, lines 1442, 544, 550, 621,732, (lines 954 and 959 the sign of degrees Celsius is not the same). In line 49 of the introduction there is mention of among all inorganic perovskites, please be more specific which are these perovskites and why the one selected in this review paper is better. In lines 68 and 69, the authors mention combining perovskites with suitable/proper nanomaterials, please add references and mention a few of them.

Author Response”

As per the reviewer’s recommendation, those typo errors were rectified. The need for this review is already justified in the introduction section with an additional paragraph with enhanced references.

Almost all of the sensors mentioned are based on PL measurements, please mention in the introduction if Cesium Lead Halide perovskites have been used for the development of other type of sensors, electrochemical etc. and add the necessary references.

Author Response”

We do respect the reviewer's suggestion. But we have already delivered the general statement to cover all those sensors in lines 50 - 54.

“Among all inorganic perovskites, the cesium lead halides (CsPbX3; X = Cl, Br and I) and composites were widely demonstrated in photovoltaic applications and in detection/quantification of metal ions, anions, chemicals, explosives, bio-analytes, pesticides, fungicides, cellular imaging, volatile organic compounds (VOC), toxic gases, humidity, temperature, and radiation [14 -19]”

The reviewer must be aware that “volatile organic compounds (VOC), toxic gases, humidity, temperature, and radiation” have been covered by photoresistive and chemoresistive sensors. Thus, at this stage, more details given on those sensors are not necessary.

  In section 3 please do not start by showing a figure. It should be placed later on.

Author Response”

As per the reviewer’s recommendation, Figure 3 is moved after the text statement.

Generally speaking when you describe the sensors developed by many researchers, do not just mention what they did, be more critical and comment on these. Try to comment on the mechanism or why a mechanism of sensing is not given. What are the problems with selectivity stability etc. and studies in real samples and how the researches did overcome them. By going from one sensor to the other, especially if they are developed for the same analyte, try to compare them.

Author Response”

Thanks for your valuable question. The scope of this review is to emphasize the importance of CsPbX3-based sensors, and the progress with underlying certain mechanisms. For comparison, we have also delivered tables. Note that not all the probes were demonstrated subsequently. Many of them were almost reported at the same time. Thus, delivering the review in chronological order as suggested by the reviewer is not possible at this stage. The sensing mechanism is described wherever required. In some reports, the underlying mechanisms were not given or discussed by the authors, which are mentioned in this review.

In table one most of the sensors listed there are for copper ions.  Also, the method used for the synthesis of the perovskite is most of the times hot injection. Try first to explain why such an interest for these particular metal ions and especially copper and also which methods have been used mostly for the synthesis of the perovskites and what are the advantages of each method.

Author Response”

Thanks for your valuable question. The scope of this review is to state the importance of CsPbX3-based sensors, and the progress with certain mechanisms. Advantages and limitations are already delivered in section 10. To address the above question, we are currently preparing an in-depth critical review. Adding more details may affect the readability/citation metrics of our review. We thank reviewer’s understanding.

Reviewer 3 Report

Comments and Suggestions for Authors

- The humidity influence on perovskite materials characteristics should be better highlighted. On the one hand, humidity can significantly disrupt the electrical and optical characteristics. But on the other hand, this significant influence can be useful in the case of humidity sensors (see https://doi.org/10.3390/ma15238369).

- Also, the temperature influence on perovskite material parameters must be better synthesized. This influence is presented fragmentarily in the manuscript, but not synthetically and conclusively.

- It should be mentioned about the chemical and physical stability of Cesium Lead Halide Perovskites compared to other perovskites materials. This aspect is important from the perspective of their applicability in the area of sensors.

Comments on the Quality of English Language

..

Author Response

Reviewer 3

We are grateful to the reviewer for the valuable comment and appreciation.

- The humidity influence on perovskite materials characteristics should be better highlighted. On the one hand, humidity can significantly disrupt the electrical and optical characteristics. But on the other hand, this significant influence can be useful in the case of humidity sensors (see https://doi.org/10.3390/ma15238369).

“Author Response”

We agree with the reviewer, the following sentences are included to state the importance of Cesium lead halides-based humidity sensors.

“Humidity and moisture are the main causes resulting to perovskite material degradation, which affect the practical use and commercialization of the perovskite-based energy devices [136]. The water molecules in air react with the metal halide perovskite surface, which rapidly affects the morphology and uniformity resulting to changes in optical properties and conductivity [137]. The above effect is the major mechanism of humidity sensing responses. Doping with specified metal ions can improve the environmental stability of metal halide perovskites and reduce the humidity effect on surface/morphology [138].”

 Also, the temperature influence on perovskite material parameters must be better synthesized. This influence is presented fragmentarily in the manuscript, but not synthetically and conclusively.

“Author Response”

As per suggestion, the following sentences are included to state the importance of Cesium lead halides-based temperature sensors in lines 1263 -1268.

“Variations in temperature led to changes in phase and grain sizes [142], which can be adopted in the temperature sensors via monitoring changes in the I-V responses, PL intensity, absorbance, etc. On the other hand, alterations in temperature also affected the grain uniformity and morphologies of CsPbX3 and composites [143] resulting to changes in PL intensity and current density. However, doping of metal ions may also improve the temperature sensitivity of CsPbX3 and composites [144] as illustrated in this section.”

It should be mentioned about the chemical and physical stability of Cesium Lead Halide Perovskites compared to other perovskites materials. This aspect is important from the perspective of their applicability in the area of sensors.

“Author Response”

As per recommendation, the following sentences are included in section 2 for comparing the stability of Cesium Lead Halide Perovskites compared to other Perovskite materials with proper citation.

“The stability of hybrid halide perovskites also follows a similar trend as the CsPbX3-based materials [9]. However, the metal oxide perovskites show slightly better stability than that of the CsPbX3 and hybrid halide perovskites [9 –12].”

Reviewer 4 Report

Comments and Suggestions for Authors

This is a nice review on the cesium lead halide perovskites for sensor applications.  Researches on various analytes are summarized.  I feel two problems.

1. The word "conjugate" is not suitable in analytical chemistry because we anticipate something related with pi-conjugation in chemistry.

2. The explanations about the mechanism of sensing of each analyte are too brief.  

If those problems are amended, I feel this can be published as a review.

Comments on the Quality of English Language

I detect no problem.

Author Response

Reviewer 4

This is a nice review on the cesium lead halide perovskites for sensor applications.  Researches on various analytes are summarized.  I feel two problems.

We are very grateful to the reviewer for providing the opportunity to improve the standard of this review article.

  1. The word "conjugate" is not suitable in analytical chemistry because we anticipate something related with pi-conjugation in chemistry.

“Author Response”

We replace the word “Conjugate” with “Composites”.

  1. The explanations about the mechanism of sensing of each analyte are too brief.  If those problems are amended, I feel this can be published as a review.

“Author Response”

The mechanisms for sensors are provided wherever it is required. Further addition/discussion on the mechanism may affect the readability. Thus, at this stage, it is not possible to include more discussion on mechanisms.

Round 2

Reviewer 2 Report

Comments and Suggestions for Authors

The authors replied to all my comments and suggestions, but I don’t agree with most of their replies and the changes they did in a very short time are very limited, they didn’t improve the quality of this very long review.

The review is very long and very specific on a specific type of perovskite for too many applications. That way no proper comparison with other perovskites developed for the same applications is made and the way this review is written is by just simply presenting the results of papers and of sensors developed without critically commenting on them.

The English in the whole manuscript is ok, however in general it could be improved. The words sensing and sensor are still wrongly used in lines 97 and 945. Please check the whole manuscript again carefully.

The keyword biosensor is deleted but replaced by “bioanalytes quantitation”, this expression has never been used as keyword in any paper. Maybe they could use quantitative determination of bioanalytical molecules but it quite long. Something else should be used.

The statement the authors did, that by mentioning the stability is sections 8 and 9, this will affect the readability/citation metrics of the review ! I cannot understand. In particular, in sections 8 and 9 where all measurements for biomolecules should be made only in aqueous solutions, the stability of the perovskite in water should be emphasized.

In addition. in section 8 the selectivity issue of the perovskite sensor is important and there in no discussion about that.

Regarding my comment if other types of sensors have been developed using this particular perovskite, I don’t mean for different analytes the authors mention, but types of sensors, electrochemical, thermal, mass, chemoresistive etc. There is no mention about that, neither why the optical sensors developed are more important based on the properties of the perovskite.

There is no discussion why these analytes are chosen by many groups to develop sensors for them and why a perovskite sensor would be better than other sensors that exist for these analytes.

Comments on the Quality of English Language

The English in the whole manuscript is ok, however in general it could be improved. The words sensing and sensor are still wrongly used in lines 97 and 945. Please check the whole manuscript again carefully.

Author Response

Response to the Reviewer’s Comments

The authors replied to all my comments and suggestions, but I don’t agree with most of their replies and the changes they did in a very short time are very limited, they didn’t improve the quality of this very long review.

We are very grateful to the reviewer for providing one more opportunity to improve the standard of this review article.

The review is very long and very specific on a specific type of perovskite for too many applications. That way no proper comparison with other perovskites developed for the same applications is made and the way this review is written is by just simply presenting the results of papers and of sensors developed without critically commenting on them.

“Author Response”

As suggested by the reviewer, we have added extra sections to comment applications in various aspects as following.

3.1. Critical comments on CsPbX3 (X = Cl, Br, and I)-based metal ions detection  

Based on the existing results, it is noted that the as-synthesized CsPbX3 QDs, NCs, and NWs display high specific selectivity to Cu2+ through feasible energy transfer between the probes and Cu2+ [80–82]. Furthermore, it was clarified that the Yb3+ doping enhanced the selectivity by reducing the surface defect [80, 81], thereby suggesting the effectiveness of surface forces in sensors. Another critical issue of the use of CsPbX3 (X = Cl, Br, and I) probes for metal ions detection, which requires more attention, is their stability in aquatic environments. To solve the stability issues, polymers, and ligand capping/coating over CsPbX3 were proposed [83, 84, 86, 89, and 90], which may enhance the PLQY by avoiding surface exposure to environmental forces existing in water and air. However, whether this approach can can be effective when exposeing to Cu2+ in an aquatic environment still remains an oprn quection. The development of the pass filter consisting of CsPbBr3 and CsPb(Cl/Br)3 QDs was demonstrated for Cu2+ detection via PL quenching responses [85]. However, development of such pass filters has not meet the commercial standard. It is a premature proposal and requires additional work. The CsPbBr3 QDs/CsPbI3 QDs were explored in the microfluidic tactic facilitated detection of Cu2+ [87, 88]. However, fabrication processes of such devices are rather complicate and require well-equipped clean room environment, thereby restricting their advance in most developing countries. Also, it is essential to determine whether this microfluidic tactic is effective in all environmental samples. The use of CsPbBr3crystals, NCs, and QDs was also reported in PL “Turn-Off” detection of Hg2+ and UO22+ [92–94,97]. Among the available reports on CsPbX3 (X = Cl, Br, and I)-based metal ion sensors, many of them confirmed their selectivity to Cu2+, however, the underlying mechanisms of detecting Hg2+ and UO22+ by CsPbBr3 crystals, NCs, and QDs are still unclear. Likewise, composites of CsPbX3 (X = Cl, Br, and I) with other emerging nanomaterials, such as MOFs, Mxene, APTES, etc., have been proved to be effective in discriminating diverse heavy metal ions [91,94–96,98–100]. However, most of those reports did not address the feasible surface-facilitated detection mechanisms, which restricted the development of analytical devices. These results also raise the question on the reliability of CsPbX3 (X = Cl, Br, and I)-based Cu2+ sensors. The reason behind the selective sensing of Cu2+ must be clarified by investigating the Pb2+replacement mechanism, as well as the magnetic property (ferro/ferri-electronic) changes. Crystalline and lattice features of the probes/compositions in the presence/absence of analytes are not considered from mechanistic aspects, which should be taken into account the sensor designs in future. If the crystalline/lattice parameters of CsPbX3 (X = Cl, Br, and I)-based probes were taken into account, it is highly feasible to design chemoresistive and electrochemical sensors for heavy metal quantification in real samples.  

4.1. Critical view on CsPbX3 (X = Cl, Br, and I)-based anion sensors  

CsPbX3 (X = Cl, Br, and I)-based probes/composites have been reported for discriminating Cl-, Br- and I- to display red/blue shifted PL emissive peaks via the anion exchange as noted in the Table 2 [101–112]. Among these sensing studies, if the anions are present in the aquatic environment, this may also affect the rapid anion exchange due to the disturbed structural parameters. These issues should be addressed with in depth investigations. Surface stabilization by using suitable capping agents may change the lattice features resulting to enhanced PLQY, thus the proposed anion sensing performance by CsPbX3 (X = Cl, Br, and I)-based probes/composites is not yet confirmed and requires further interrogations. The real question is, how can one believe the anion sensing performance if the sensing media itself could affect the stability of the proposed CsPbX3 (X = Cl, Br, and I)-based probes/composites. The reaction-based sensing of S2- was also witnessed by the tetraphenylporphyrin tetrasulfonic acid (TPPS) modified CsPbBr3 NCs [113], which showed dependence on the composition concentrations. Thereby, optimization of the composition concentrations is regarded as a high-priority task and requires detailed interrogations. Due to the instability issues of CsPbX3 (X = Cl, Br, and I)-based probes, anion discrimination to distinct competing matrices in real water samples becomes more concerned.    

5.1. Critical view on CsPbX3 (X = Cl, Br, and I)-based chemicals and explosive sensors   

Detection/quantification of specialized chemicals, such as CH3I, CH2Cl2, CH2Br2, benzoyl peroxide, and excessive acid number (AN) via the anion exchange mechanisms [114 -121] cannot be regarded as a specific quantification procedure because of its similarity to the anions detection. This should be critically investigated to pursue the “state-of-the-art” sensing procedure. Since the observed ratiometric PL responses are also similar to those in the anions sensing studies, critical investigations are required for commercialization. Furthermore, discriminating explosives was demonstrated via the PL quenching response resulted from the surface interaction and charge transfer between nitro-containing explosives and CsPbX3 (X = Cl, Br, and I)-based probes or composites [122–124]. However, this also requires critical studies to justify the exact static/dynamic PL quenching responses.

6.1. Critical view on CsPbX3 (X = Cl, Br, and I)-based gases and VOCs detection   

When combining the CsPbX3 (X = Cl, Br, and I) with different materials, it can result to composited materials with exceptional electro-optical properties and less defect, which can be adopted in the design of PL-based probes, electrochemiluminescence probes, chemoresistive sensors for discriminating N2, O2, H2S, tripropylamine (TPrA), thiophene sulfides, and NH3 [125–150]. To achieve the above goal the compositing ratios need to be optimized. Changes in the compositing ratios may significantly affect the selectivity, thereby requiring careful/critical adjustments. The detection of H2S was demonstrated in the Rat brain and zebrafish studies [133–135], but there is no clear indication of how to overcome the toxicity induced by Pb2+ in CsPbX3. The film- or test strip-based sensing of NH3mostly displayed dependence on the crystalline and morphological features of CsPbX3 (X = Cl, Br, and I) and composites. Thus, optimizing the crystallinity/morphology of thin film is critical to attain the best results.

7.1. Critical view on CsPbX3 (X = Cl, Br, and I)-based Humidity, temperature, and radiation/photodetection

The low stability and degradation of CsPbX3 (X = Cl, Br, and I) and composites were adopted as the sensor responses for humidity detection [151–156]. However, reports on the humidity sensors also talked about the recovery, which were problematic due to the environmental instability issue of the CsPbX3 (X = Cl, Br, and I) and composites. Thus, CsPbX3(X = Cl, Br, and I) and composites-based recovery of humidity sensors must be carefully examined. Phase transitions may occurred in the CsPbX3 (X = Cl, Br, and I) and composites during the temperature sensing [157–167], thus in depth investigations are critically in many cases involving the phase transitions. Likewise, doping of ions, such as Mn2+may enhance the phase changes, which requires a more careful examination. When detecting photon/radiation, lattice defects could be generated in the CsPbX3 (X = Cl, Br, and I). The situation can be alleviated by using composites with diverse materials [168–180]. However, the compositing ratios must be critically evaluated to achieve better sensing results.

8.1. Critical view on CsPbX3 (X = Cl, Br, and I)-based bio-analytes, drugs, fungicides, and pesticide discrimination

Stability of CsPbX3 (X = Cl, Br, and I) in water is the main issue for discriminating bio-analytes, drugs, fungicides, and pesticides [181–210]. To avoid above complications, compositing CsPbX3 (X = Cl, Br, and I) with other materials, such as APTES, BN, and MIPs have been proposed. However, there are still a few probes require critical justifications for structural degradation in aqueous media. The water-stable CsPbBr3/Cs4PbBr6 NCs was proposed for detecting folic acid [202], but how long can the proposed structure remained stable is still under debate. Another controversial issue on the CsPbX3 (X = Cl, Br, and I) and composites-based biomolecule sensing is how to avoid toxicity of Pb2+ toward food sample analysis. Consequently, the sensor selectivity to specific analytes requires careful investigations in many reports. Detecting pesticides and fungicides in real samples using CsPbX3 (X = Cl, Br, and I) and composites needs critical interrogations to overcome toxicity induced by Pb2+.  

9.1. Critical view on CsPbX3 (X = Cl, Br, and I)-based cellular imaging

It has been argued that cellular imaging using the CsPbX3 (X = Cl, Br, and I) could be hindered due to the unstable emission properties of CsPbX3 in aqueous media. To avoid the instability issue, capping CsPbX3 (X = Cl, Br, and I) with suitable ligands, such as 3-amino-propyl)trimethoxysilane (APTMS), polyethylene glycol grafted phospholipid (mPEG-DSPE), and alkoxysilanes (TMOS and phTEOS) have been proposed. This capping could not only reduces toxicity due to Pb2+ but also improves biocompatibility [211-215], however, careful optimization is still required. Polymer capping, metal doping, and surface coating were also engaged to improve the biocompatibility of CsPbX3 (X = Cl, Br, and I) in cellular imaging studies [216–221]. Nevertheless, critical assessments of the toxic profiles when applying those composites in cellular imaging still require much attention.  

The English in the whole manuscript is ok, however in general it could be improved. The words sensing and sensor are still wrongly used in lines 97 and 945. Please check the whole manuscript again carefully.

“Author Response”

As suggested by the reviewer, we have enhanced the English of the whole manuscript. The words sensing and sensors are placed wherever required for precise meaning.

The keyword biosensor is deleted but replaced by “bioanalytes quantitation”, this expression has never been used as keyword in any paper. Maybe they could use quantitative determination of bioanalytical molecules but it quite long. Something else should be used.

“Author Response”

We agree with the reviewer. The Keyword “Bioanalaytes detection” has been replaced by “Environmental monitoring”.

The statement the authors did, that by mentioning the stability is sections 8 and 9, this will affect the readability/citation metrics of the review! I cannot understand. In particular, in sections 8 and 9 where all measurements for biomolecules should be made only in aqueous solutions, the stability of the perovskite in water should be emphasized. In addition. in section 8 the selectivity issue of the perovskite sensor is important and there in no discussion about that.

“Author Response”

To address the above issues, the following critical comments were included at the end of sections 8 and 9.

8.1. Critical view on CsPbX3 (X = Cl, Br, and I)-based bio-analytes, drugs, fungicides, and pesticide discrimination

Stability of CsPbX3 (X = Cl, Br, and I) in water is the main issue for discriminating bio-analytes, drugs, fungicides, and pesticides [181–210]. To avoid above complications, compositing CsPbX3 (X = Cl, Br, and I) with other materials, such as APTES, BN, and MIPs have been proposed. However, there are still a few probes require critical justifications for structural degradation in aqueous media. The water-stable CsPbBr3/Cs4PbBr6 NCs was proposed for detecting folic acid [202], but how long can the proposed structure remained stable is still under debate. Another controversial issue on the CsPbX3 (X = Cl, Br, and I) and composites-based biomolecule sensing is how to avoid toxicity of Pb2+ toward food sample analysis. Consequently, the sensor selectivity to specific analytes requires careful investigations in many reports. Detecting pesticides and fungicides in real samples using CsPbX3 (X = Cl, Br, and I) and composites needs critical interrogations to overcome toxicity induced by Pb2+.

9.1. Critical view on CsPbX3 (X = Cl, Br, and I)-based cellular imaging

It has been argued that cellular imaging using the CsPbX3 (X = Cl, Br, and I) could be hindered due to the unstable emission properties of CsPbX3 in aqueous media. To avoid the instability issue, capping CsPbX3 (X = Cl, Br, and I) with suitable ligands, such as 3-amino-propyl)trimethoxysilane (APTMS), polyethylene glycol grafted phospholipid (mPEG-DSPE), and alkoxysilanes (TMOS and phTEOS) have been proposed. This capping could not only reduces toxicity due to Pb2+ but also improves biocompatibility [211-215], however, careful optimization is still required. Polymer capping, metal doping, and surface coating were also engaged to improve the biocompatibility of CsPbX3 (X = Cl, Br, and I) in cellular imaging studies [216–221]. Nevertheless, critical assessments of the toxic profiles when applying those composites in cellular imaging still require much attention.  

Regarding my comment if other types of sensors have been developed using this particular perovskite, I don’t mean for different analytes the authors mention, but types of sensors, electrochemical, thermal, mass, chemoresistive etc. There is no mention about that, neither why the optical sensors developed are more important based on the properties of the perovskite. There is no discussion why these analytes are chosen by many groups to develop sensors for them and why a perovskite sensor would be better than other sensors that exist for these analytes.

“Author Response”

To address those issues, the following paragraph has been included in the introduction section.

The exceptional optical properties, unique structural/crystalline features, and electronic structures of CsPbX3 (X = Cl, Br, and I), are considered important material properties for the electrochemical, thermal, and chemoresistive sensing studies. To date, numerous optical sensors made of CsPbX3 (X = Cl, Br, and I) and composites have been throughly investigated with exceptional applicability [9–15]. This can be attributed to their distinct and high PLQY in red to blue luminescence. However, there were reports on electrochemical, thermal, and chemoresistive sensing performance of CsPbX3 (X = Cl, Br, and I) and composites [9–15], which require further clarification view for future research. Heavy metal ions and anions are well-known environmental contaminants involving in cellular processes and, at high concentrations, they may become harmful to living beings as well [44–46]. Chemicals and explosives also contaminate the environment; thus, their detection methods are available in many reports [47–49]. Toxic gases and VOCs are noted as vital industrial contaminants, thus their quantitation has been explored by numerous researchers [50,51]. Exposure to radiation, temperature, and high humidity may harm living tissues and beings, thus researchers developed sensors for photo, radiation, and photo-detection [52–54]. Bio-analytes, drugs, fungicides, and pesticide play crucial roles in food cycles and sustain the living environment, thus numerous reports were available for their identification [55–58]. Based on the aforementioned important issues, many researchers adopted the CsPbX3 (X = Cl, Br, and I) and composites for optical, electrochemical, chemoresistive, and thermal detection of those analytes. The progress and challenges in developing these sensors are reviewed in this article.